# Regularized least squares learning
# with heavy-tailed noise is minimax optimal

**Mattes Mollenhauer**
Merantix Momentum
mattes.mollenhauer@merantix-momentum.com

**Nicole Mücke**
Technische Universität Braunschweig
nicole.muecke@tu-braunschweig.de

**Dimitri Meunier**
Gatsby Computational Neuroscience Unit, UCL
dimitri.meunier.21@ucl.ac.uk

**Arthur Gretton**
Gatsby Computational Neuroscience Unit, UCL
arthur.gretton@gmail.com

## Abstract

This paper examines the performance of ridge regression in reproducing kernel Hilbert spaces in the presence of noise that exhibits a finite number of higher moments. We establish excess risk bounds consisting of subgaussian and polynomial terms based on the well known integral operator framework. The dominant subgaussian component allows to achieve convergence rates that have previously only been derived under subexponential noise—a prevalent assumption in related work from the last two decades. These rates are optimal under standard eigenvalue decay conditions, demonstrating the asymptotic robustness of regularized least squares against heavy-tailed noise. Our derivations are based on a Fuk–Nagaev inequality for Hilbert-space valued random variables.

## 1 Introduction

Given two random variables $X$ and $Y$, we seek to empirically minimize the expected squared error

$$\mathcal{R}(f) := \mathbb{E}\big[(Y - f(X))^2\big]$$

over functions $f$ in a *reproducing kernel Hilbert space* $\mathcal{H}$ consisting of functions from a topological space $\mathcal{X}$ to $\mathbb{R}$. We consider the standard model

$$Y = f_\star(X) + \varepsilon$$

with the *regression function* $f_\star : \mathcal{X} \to \mathbb{R}$ and noise variable $\varepsilon$ satisfying $\mathbb{E}[\varepsilon|X] = 0$. Given $n$ independent sample pairs $(X_i, Y_i)$ drawn from the joint distribution of $X$ and $Y$, we investigate the classical *ridge regression estimate*

$$\widehat{f}_\alpha := \underset{f \in \mathcal{H}}{\arg\min} \frac{1}{n} \sum_{i=1}^{n} (Y_i - f(X_i))^2 + \alpha \|f\|_{\mathcal{H}}^2 \tag{1}$$

39th Conference on Neural Information Processing Systems (NeurIPS 2025).

with *regularization parameter* $\alpha > 0$. We adopt the well-known perspective going back to the pathbreaking work [1–4], which characterizes $\widehat{f}_\alpha$ as the solution of a linear inverse problem in $\mathcal{H}$ obtained by performing *Tikhonov regularization* [5] on a stochastic discretization of the integral operator induced by the kernel of $\mathcal{H}$ and the marginal distribution of $X$. Since its inception, this approach has been refined and generalized in a multitude of ways, including more general learning settings and alternative algorithms and applications. We refer the reader to [6–23] and the references therein for an overview. A common theme in the above line of work is the derivation of confidence bounds of the *excess risk*

$$\mathcal{R}(\widehat{f}_\alpha) - \mathcal{R}(f_\star) = \mathbb{E}[(\widehat{f}_\alpha(X) - f_\star(X))^2]$$

i.e., with high probability over the draw of the sample pairs under appropriate regularity assumptions about the regression function $f_\star$ and distributional assumptions about $\varepsilon$.

**Heavy-tailed noise.** In this work, we assume that the real-valued random variable $\varepsilon$ has only a finite number of higher conditional absolute moments, i.e., there exists some $q \in \mathbb{N}$, $q \geq 3$ such that

$$\mathbb{E}[|\varepsilon|^q | X] < Q < \infty \text{ almost surely.} \tag{2}$$

This setting covers noise associated with distributions without a moment generating function—for example the $t$-distribution, Fréchet distribution, Pareto distribution and Burr distribution (correspondingly centered). In such a setting, the family of *Fuk–Nagaev inequalities* [24, 25] provides sharp nontrivial tail bounds beyond Markov's inequality for sums of heavy-tailed real random variables. These results show that the tail is dominated by a subgaussian term [26] in a small deviation regime (reflecting the central limit theorem) and a polynomial term in a large deviation regime. In order to apply this fact to the integral operator approach, we modify a vector-valued version of the Fuk–Nagaev inequality going back to [27] for random variables taking values in Hilbert spaces. In a practical context, heavy-tailed noise satisfying the moment condition (2) plays a role in fields such as finance, insurance, communication networks and atmospherical sciences [28, 29].

**Prior work: Bernstein condition.** In the aforementioned context of spectral regularization algorithms in kernel learning, existing work generally assumes that $\varepsilon$ is subexponential.[1] In particular, the so-called *Bernstein condition*[2] requires the existence of constants $\sigma, Q > 0$ almost surely satisfying

$$\mathbb{E}[|\varepsilon|^q | X] \leq \frac{1}{2} q! \sigma^2 Q^{q-2} \text{ almost surely} \tag{3}$$

for all $q \geq 2$. This condition allows to apply a Hilbert space Bernstein inequality [31] to the well-known integral operator framework in order to obtain convergence results. We refer the reader to [3, 6–11, 13] for a selection of results in this setting. To our knowledge, all results obtaining optimal rates in this setting rely on the Bernstein tail bound. The importance of the Bernstein inequality in the context of this work is emphasized by the *effective dimension* [3, 32], which measures the capacity of the hypothesis space $\mathcal{H}$ relative to the choice of the regularization parameter $\alpha$ and the marginal distribution of $X$ in terms of the eigenvalues of the integral operator. When used as a variance proxy in the Bernstein inequality, the effective dimension is the central tool that allows to derive minimax optimal rates under assumptions about the eigenvalue decay, as first shown by [3] and subsequently refined in the aforementioned work. Due to this elegant connection between eigenvalue decay and concentration, the integral operator formalism has been predominantly focused around the assumption (3) over the last two decades. Similar approaches based on the Bernstein inequality with suitable variance proxies are commonly applied across a variety of estimation techniques in order to obtain fast rates [e.g. 33, 34].

**Overview of contributions.** In this work, we show that the rates derived under the Bernstein condition (3) in the mentioned literature can equivalently be obtained with the significantly less restrictive higher moment assumption (2) with the same regularization parameter schedules. We consider both the *capacity independent* setting (i.e., without assumptions about the eigenvalue decay of the integral operator [4, 6]) and the more common *capacity dependent* setting involving

---

[1]The term *subexponential* in this work refers to light-tailed distributions in the Orlicz sense [26] in contrast to alternative definitions for heavy-tailed distributions found in the literature [28].

[2]Condition (3) is in fact related to the slightly stronger *subgamma property*, see [30]. Applying Stirling's approximation to the right hand side of (3) gives the typical subexponential bound for $L^q$-norms of $\varepsilon$, see [26].

the effective dimension [e.g. 3, 11, 35]. Even though the capacity dependent results are sharper, we dedicate a separate discussion to the capacity independent setting, as it allows a less technical presentation and a simplified and insightful asymptotic dicussion. In both settings, we base our analysis on a Hilbert space version of the Fuk–Nagaev inequality, providing excess risk bounds that exhibit both subgaussian and polynomial tail components. The dominant subgaussian term allows to asymptotically recover the familiar bounds known from the subexponential noise scenario. In the capacity dependent setting, we use the effective dimension not only as variance proxy, but also as a *proxy for the higher moments occurring in the Fuk–Nagaev inequality*—the resulting bound is sharp enough so that standard assumptions about the eigenvalue decay lead to known optimal convergence rates. This technique directly generalizes the aforementioned approach based on the Bernstein inequality.

**Practical implications, future work and limitations.** The square loss is often not used in practice when one expects heavy-tailed noise, as it is sensitive to outliers when used without regularization [36]. However, when used with regularization in the presence of noise of the form (2), we show that it exhibits a certain degree of robustness. In particular,

(i) it asymptotically achieves the optimal rates with high probability known from the light-tailed setting with the same regularization schedule,

(ii) in a large sample setting, the confidence behavior of the excess risk is essentially subexponential,

(iii) in a small sample setting, the confidence behavior is polynomial and stronger regularization is required due to the impact of the heavy tails.

We focus on the original well-specified kernel ridge regression setting as investigated by [3] in order to simplify the presentation and highlight the key arguments. However, we expect our approach to transfer to other settings, for example involving more general source conditions [6, 37], general spectral filter methods [9, 20], misspecified models [11, 38], the kernel conditional mean embedding with unbounded kernels on the target space [17], high- and infinite-dimensional output spaces [15, 16] and many other settings allowing for the application of the integral operator formalism. We believe that kernel regression with unbounded kernels admitting a finite higher moment can be analyzed with a similar technical approach as the one presented here. Let us mention some limitations of the present work. While our results show a certain degree of robustness of regularized least squares against heavy-tailed noise for $q \geq 3, q \in \mathbb{N}$, we expect our results to directly transfer to all real $q > 2$, as versions of the real-valued Fuk–Nagaev bound cover this case [39, 40]. Currently, this restriction of our results exclusively depends on the validity of the Fuk–Nagaev bound in Hilbert spaces for these $q$ as an artefact of the proof technique by [27], which we modify. Furthermore, for $q < 2$, it is clear that the square loss is not necessarily well-defined and a different loss such as the Cauchy loss should be used [41]. Our results demonstrate that, when operating in a high confidence setting, heavy-tailed noise may require a significantly higher level of regularization than light-tailed noise, making empirical regularization parameter choice rules as a function of the confidence level very important. Extending the analysis of classical parameter choice rules for deterministic inverse problems [42] to the stochastic setting based on heavy-tailed noise may therefore be an interesting future direction.

**Other related work.** We are not aware of any specific analysis of kernel ridge regression and the integral operator formalism in the heavy-tailed scenario given by (2) in the literature. However, there exists a wide variety of related results for regression with heavy-tailed noise and robust estimation— we put our results in the context of the most important related work. Optimal rates for (unpenalized) least squares regression over nonparametric hypothesis spaces can generally only be derived in the empirical process context when $q$ in condition (2) is large enough with respect to suitable metric entropy requirements, see [43–46]. In comparison, our setting allows to recover optimal rates for the reproducing kernel Hilbert space scenario with a regularization schedule which is independent of $q$. For linear models over finite basis functions, [47] derives exponential concentration of the ridge estimator under finite variance on the noise for fixed $\alpha$. We also highlight the field of robust estimation techniques outside of the standard least squares context, see e.g. [36, 47–51] and the references therein. While the analysis of robust finite-dimensional linear regression under heavy-tailed noise requires discussions of the distribution of the covariates and their covariance matrix (often under variance-kurtosis equivalence [52–54]), we impose the typical assumption that the kernel is

bounded, leading to subgaussian concentration of the embedded covariates in the potentially infinite-dimensional feature space. Recently, [41] derived nearly optimal rates for kernel ridge regression with Cauchy loss under (2) with $q > 0$ depending on the Hölder continuity parameter of the target function using more classical arguments. Finally, from a more technical perspective, the approach by [55] shares similarities with the methods applied in our paper: The authors use a real-valued Fuk–Nagaev inequality to bound the stopping time complexity of stochastic gradient descent for ordinary least squares regression. However, [55] provides results only for the finite-dimensional setting based on a martingale decomposition by assuming a lower bound on the minimal eigenvalue of the covariance matrix. In contrast, our work targets the infinite-dimensional setting based on inverse problem theory without such a lower bound and explicitly proves minimax optimality.

**Structure of this paper.** We introduce our notation and basic preliminaries in Section 2. In Section 3, we provide the excess risk bound for the capacity independent setting and derive corresponding rates. Section 4 contains an excess risk bound based on the effective dimension and recovers rates which are known to be minimax optimal also in the subexponential noise setting. Finally, in Section 5, we briefly discuss the Fuk–Nagaev inequality used to derive our results. Appendix A contains a numerical experiment which confirms the behavior of the excess risk described by our theoretical results. We report all proofs for the results in the main text in Appendix B and provide additional technical results as individual appendices.

## 2 Preliminaries

We assume that the reader is familiar with the analysis of linear Hilbert space operators [56, 57] and the basic theory of reproducing kernel Hilbert spaces [34, 58]. Let $\pi$ denote the marginal distribution of $X$ and $L^2(\pi)$ denote the space of real-valued Lebesgue square integrable functions with respect to $\pi$. We write $\mathcal{L}(H_1, H_2)$ for the space of bounded linear operators between Hilbert spaces $H_1$ and $H_2$ with operator norm $\|\cdot\|$ and abbreviate $\mathcal{L}(H_1) = \mathcal{L}(H_1, H_1)$. We additionally consider the space of *Hilbert–Schmidt operators* $\mathcal{S}_2(H_1, H_2) \subset \mathcal{L}(H_1, H_2)$ and the space of *trace class operators* $\mathcal{S}_1(H_1, H_2) \subset \mathcal{S}_2(H_1, H_2)$ with norms $\|\cdot\|_{\mathcal{S}_1(H_1, H_2)}$ and $\|\cdot\|_{\mathcal{S}_2(H_1, H_2)}$ and the *trace* $\mathrm{tr}(\cdot)$. The *adjoint* of $A \in \mathcal{L}(H_1, H_2)$ is written as $A^* \in \mathcal{L}(H_2, H_1)$.

### 2.1 Reproducing kernel Hilbert space

We consider the *reproducing kernel Hilbert space* (RKHS) $\mathcal{H}$ consisting of functions from $\mathcal{X}$ to $\mathbb{R}$ induced by the *symmetric positive semidefinite kernel* $k \colon \mathcal{X} \times \mathcal{X} \to \mathbb{R}$. with *canonical feature map*

$$\phi(x) : \mathcal{X} \to \mathcal{H},$$
$$x \mapsto k(\cdot, x),$$

i.e. we have the *reproducing property* $f(x) = \langle f, \phi(x) \rangle_{\mathcal{H}}$ for all $x \in \mathcal{X}$ and $f \in \mathcal{H}$.

**Assumption 2.1** (Domain and kernel)**.** We impose the following standard assumptions throughout this paper in order to avoid issues related to measurability and integrability [34, Section 4.3]:

   (i) $\mathcal{X}$ is a second-countable locally compact Hausdorff space, $\mathcal{H}$ is separable (this is satisfied if $k$ is continuous, given that $\mathcal{X}$ is separable),

  (ii) $k(\cdot, X)$ is almost surely measurable in its first argument,

 (iii) $k(X, X) \leq \kappa^2$ almost surely for some finite constant $\kappa$.

All assumptions above hold for commonly used continuous radial kernels on $\mathbb{R}^d$ such as the Gaussian kernel and the Matèrn kernel. However, the boundedness assumption is violated for polynomial kernels unless $X$ is bounded almost surely.

**Integral and covariance operators.** Under Assumption 2.1, we may consider the typical linear operators associated with $\pi$ and $k$. We consider the *embedding operator*

$$I_\pi \colon \mathcal{H} \hookrightarrow L^2(\pi),$$
$$f \mapsto [f]_\pi$$

identifying $f \in \mathcal{H}$ with its equivalence class $[f]_\pi \in L^2(\pi)$. The adjoint $I_\pi^* : L^2(\pi) \to \mathcal{H}$ is given by

$$I_\pi^* f = \int_{\mathcal{X}} \phi(x) f(x) \, \mathrm{d}\pi(x) \in \mathcal{H}, \quad f \in L^2(\pi).$$

We obtain the self-adjoint *integral operator* $\mathcal{T}_\pi := I_\pi I_\pi^* : L^2(\pi) \to L^2(\pi)$ induced by $k$ and $\pi$ as

$$\mathcal{T}_\pi f = \int_{\mathcal{X}} k(\cdot, x) f(x) \, \mathrm{d}\pi(x), \quad f \in L^2(\pi).$$

The self-adjoint *kernel covariance operator* $\mathcal{C}_\pi := I_\pi^* I_\pi : \mathcal{H} \to \mathcal{H}$ is given by

$$\mathcal{C}_\pi = \int_{\mathcal{X}} \phi(x) \otimes \phi(x) \, \mathrm{d}\pi(x).$$

Assumption 2.1(iii) ensures that we have $\|I_\pi\| = \|I_\pi^*\| \le \kappa$ as well as $\|\mathcal{T}_\pi\| = \|\mathcal{C}_\pi\| \le \kappa^2$. Moreover, the operators $I_\pi$ and $I_\pi^*$ are Hilbert–Schmidt and both $\mathcal{T}_\pi$ and $\mathcal{C}_\pi$ are therefore self-adjoint positive semidefinite and trace class. By the *polar decomposition* of $I_\pi$ and $I_\pi^*$, there exist partial isometries $U : \mathcal{H} \to L^2(\pi)$ and $\tilde{U} : L^2(\pi) \to \mathcal{H}$ such that

$$I_\pi = U(I_\pi^* I_\pi)^{1/2} = U\mathcal{C}_\pi^{1/2} \text{ and } I_\pi^* = \tilde{U}(I_\pi I_\pi^*)^{1/2} = \tilde{U}\mathcal{T}_\pi^{1/2}. \tag{4}$$

In particular, we have $\|[f]_\pi\|_{L^2(\pi)} = \|I_\pi f\|_{L^2(\pi)} = \|\mathcal{C}^{1/2} f\|_{\mathcal{H}}$ for all $f \in \mathcal{H}$. We will also frequently use the fact that Assumption 2.1(iii) implies $\sup_{x \in \mathcal{X}} |f(x)| \le \kappa \|f\|_{\mathcal{H}}$.

## 2.2 Ridge regression

We introduce the standard integral operator formalism for ridge regression in RKHSs, see e.g. [3]. We consider the standard $L^2(\mathbb{P})$-orthogonal decomposition of $Y$ with respect to the closed subspace $L^2(\mathbb{P}, \sigma(X)) \subset L_2(\mathbb{P})$ of $\sigma(X)$-measurable functions given by

$$Y = f_\star(X) + \varepsilon \tag{5}$$

with the *regression function* $f_\star(X) = \mathbb{E}[Y|X] \in L_2(\mathbb{P})$ and noise variable $\varepsilon \in L^2(\mathbb{P})$ satisfying $\mathbb{E}[\varepsilon|X] = 0$. Based on the representation (5), we have $\mathcal{R}(f) = \|f - f_\star\|_{L^2(\pi)}^2 + \|\varepsilon\|_{L^2(\mathbb{P})}^2$ for all $f \in L^2(\pi)$ and hence the excess risk satisfies $\mathcal{R}(f) - \mathcal{R}(f_\star) = \|f - f_\star\|_{L^2(\pi)}^2$.

**Regularized population solution.** We define the *regularized population solution*

$$f_\alpha := \underset{f \in \mathcal{H}}{\arg\min}\, \mathbb{E}[(Y - f(X))^2] + \alpha \|f\|_{\mathcal{H}}^2 = \underset{f \in \mathcal{H}}{\arg\min}\, \|I_\pi f - f_\star\|_{L^2(\pi)}^2 + \alpha \|f\|_{\mathcal{H}}^2,$$

with $\alpha > 0$, which is alternatively expressed as

$$f_\alpha = (\mathcal{C}_\pi + \alpha \operatorname{Id}_{\mathcal{H}})^{-1} I_\pi^* f_\star \in \mathcal{H} \tag{6}$$

with the identity operator $\operatorname{Id}_{\mathcal{H}}$ on $\mathcal{H}$.

**Regularized empirical solution.** We consider sample pairs $(X_1, Y_1), \ldots, (X_n, Y_n) \sim \mathcal{L}(X, Y)$ independently obtained from the joint distribution of $X$ and $Y$. We define the empirical versions of the operators above in terms of

$$\widehat{I}_\pi^* f := \frac{1}{n} \sum_{i=1}^{n} \phi(X_i) f(X_i) \in \mathcal{H}, \quad f \in L^2(\pi)$$

as well as

$$\widehat{\mathcal{C}}_\pi := \frac{1}{n} \sum_{i=1}^{n} \phi(X_i) \otimes \phi(X_i).$$

The *empirical solution* of the learning problem in $\mathcal{H}$ with regularization parameter $\alpha > 0$ is given by the empirical analogue of (6), which we obtain in terms of

$$\widehat{f}_\alpha = (\widehat{\mathcal{C}}_\pi + \alpha \operatorname{Id}_{\mathcal{H}})^{-1} \widehat{\Upsilon} \in \mathcal{H}, \tag{7}$$

where the empirical right hand side $\widehat{\Upsilon} \in \mathcal{H}$ of the inverse problem is given by

$$\widehat{\Upsilon} := \frac{1}{n} \sum_{i=1}^{n} \phi(X_i) Y_i = \widehat{I}_\pi^* f_\star + \frac{1}{n} \sum_{i=1}^{n} \phi(X_i) \varepsilon_i. \tag{8}$$

Here, we use the orthogonal decomposition $Y_i = f_\star(X_i) + \varepsilon_i$ in the second equivalence. We note that $\widehat{\Upsilon}$ directly serves as an empirically evaluable unbiased estimate of $I_\pi^* f_\star$, as $\widehat{I}_\pi^* f_\star$ itself cannot be empirically evaluated because $f_\star$ is unknown. As usual, we interpret the above objects as random variables depending on the product measure $\mathbb{P}^{\otimes n}$ through their definition based on the observation pairs $(X_i, Y_i)$. In practice, the empirical solution $\widehat{f}_\alpha$ can be evaluated in terms of the classical *representer theorem*, see e.g. [1, Proposition 8].

## 2.3  Distributional assumptions

We list the assumptions we impose upon the distributions of $Y$, $X$ and $\varepsilon$.

**Assumption 2.2** (Moment condition). We consider the model given by (5) and assume that we have almost surely

$$\mathbb{E}[\varepsilon|X] = 0, \quad \mathbb{E}\big[\varepsilon^2|X\big] < \sigma^2 \quad \text{and} \quad \mathbb{E}[|\varepsilon|^q|X] < Q, \tag{MOM}$$

for some constants $\sigma^2 > 0$, $Q > 0$ and $q \in \mathbb{N}$, $q \geq 3$.

We now introduce a classical smoothness assumption in terms of a *Hölder source condition* [59].

**Assumption 2.3** (Source condition). We define the set $\Omega(\nu, R) := \{\mathcal{T}_\pi^\nu f \mid \|f\|_{L^2(\pi)} \leq R\} \subset L^2(\pi)$ and assume

$$f_\star \in \Omega(\nu, R) \tag{SRC}$$

for some smoothness parameter $\nu \geq 1/2$ and $R > 0$.

We give the definition of the source set $\Omega(\nu, R)$ with respect to $L^2(\pi)$ and not with respect to $\mathcal{H}$, which is also commonly found in the literature. Furthermore, the source condition is sometimes described in terms of so-called *interpolation spaces* or *Hilbert scales*. Our definition can equivalently be expressed in terms of these concepts by appropriately reparametrizing $\nu$, see e.g. [6, 11, 59] for more details.

**Remark 2.4** (Well-specified case). In this work, we explicitly consider the condition $\nu \geq 1/2$, which implies the *well-specified* setting in which we have $\Omega(\nu, R) \subset I_\pi(\mathcal{H})$. We note the case $0 \leq \nu < 1/2$ covers the *misspecified* setting, in which $\Omega(\nu, R)$ is allowed to contain elements from $L^2(\pi) \setminus I_\pi(\mathcal{H})$. We expect our approach to transfer to the misspecified setting by combining it with recent technical arguments from the literature which are outside the scope of this work [11, 15, 16, 38, 60].

## 3  Capacity-free excess risk bound

We now provide an excess risk bound and corresponding rates for kernel ridge regression in the heavy-tailed noise setting without additional assumptions about the eigenvalue decay of $\mathcal{T}_\pi$. We present this setting separately from the capacity-based results in the next section, as it allows for a clearer comparison with bounds based on subexponential noise and a simplified asymptotic discussion.

**Proposition 3.1** (Main excess risk bound). *Let* (MOM) *and* (SRC) *be satisfied. For all $\delta \in (0, 1)$ and $n \in \mathbb{N}$ such that*

$$C_\kappa \ \log(6/\delta) \leq \alpha\sqrt{n}, \qquad C_\kappa := 2(1 + \sqrt{\kappa}) \cdot \max\{1, \kappa^2\}, \tag{9}$$

*we have*

$$\|I_\pi \widehat{f}_\alpha - f_\star\|_{L^2(\pi)} \leq R\alpha^{\min\{\nu, 1\}} + \frac{C_\diamond}{\sqrt{\alpha}} \left( \frac{\log(6/\delta)}{n} + \sqrt{\frac{\alpha^{2\min\{\nu, 1\}} \log(6/\delta)}{n}} + \eta(\delta, n) \right),$$

*with confidence $1 - \delta$, with*

$$\eta(\delta, n) := \max\left\{ \left( \frac{Q}{\delta n^{q-1}} \right)^{1/q}, \sigma \sqrt{\frac{\log(6c_1/\delta)}{n}} \right\},$$

*where $0 < C_\diamond$ is given in (29) and $c_1 \geq 1$ is the constant from Proposition 5.1 depending only on $q$.*

Just as in the light-tailed setting, Proposition 3.1 shows that the optimal excess risk is achieved by balancing the contributions of the *approximation error* (e.g. the model *bias*) and *sample error* (e.g. the model *variance*) by choosing a suitable regularization parameter $\alpha$ depending on $n$ and $\delta$. The term $R\alpha^{\min\{\nu,1\}}$ quantifies the approximation error based on the smoothness of $f_\star$ and exhibits the typical *saturation effect* of ridge regression: the fact that the convergence speed cannot be improved beyond a smoothness level $\nu = 1$ [e.g. 61, 62]. The key difference to known results for subexponential noise in this setting [4, 6] is the Fuk–Nagaev term $\eta(\delta, n)$ appearing in the sample error, which introduces an additional polynomial dependence on $\delta$ and $n$. We now investigate the consequences of this term.

**Confidence regimes.** We split the confidence scale into two disjoint intervals depending on whether the subgaussian component or the polynomial component dominates in the term $\eta(\delta, n)$. For $n \in \mathbb{N}$, $q \geq 3$ we define

$$D_1(n, q) := \left\{ \delta \in (0, 1) \; : \; \eta(\delta, n) = \sigma \sqrt{\frac{\log(6c_1/\delta)}{n}} \right\}$$

$$= \left\{ \delta \in (0, 1) \; : \; n \geq \left( \frac{Q^2}{\sigma^{2q}} \right)^{\frac{1}{q-2}} \delta^{-\frac{2}{q-2}} \cdot \log(6c_1/\delta)^{-\frac{q}{q-2}} \right\}, \tag{10}$$

$$D_2(n, q) := (0, 1) \setminus D_1(n, q) . \tag{11}$$

In what follows, we will refer to $D_1(n, q)$ as the *subgaussian confidence regime* and $D_2(n, q)$ as the *polynomial confidence regime*. The *effective sample size* $n_0$ ensuring subgaussian behavior of $\eta(n, \delta)$ for all $n \geq n_0$ is hence

$$n_0 := \left( \frac{Q^2}{\sigma^{2q}} \right)^{\frac{1}{q-2}} \delta^{-\frac{2}{q-2}} \cdot \log(6c_1/\delta)^{-\frac{q}{q-2}} . \tag{12}$$

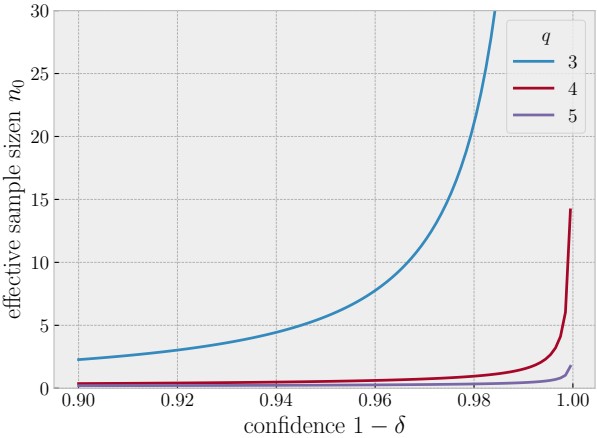

Figure 1: Illustration of the the effective sample size $n_0$ ensuring subgaussian behavior of the term $\eta(\delta, n)$ defined in (12) for different choices of $q$. For simplicity, we set $c_1 = 1$, $\sigma = 2$ and $Q = 10$.

We illustrate the behavior of $n_0$ depending on $1 - \delta$ for different choices of $q$ in Figure 1 (we note that the involved constant $c_1$ stems from the Fuk–Nagaev inequality given in Proposition 5.1 and generally depends on $q$). We choose $c_1 = 1$ for simplicity to provide a basic intuition—note that $c_1$ only affects $n_0$ logarithmically. We refer the reader to [40] for a detailed discussion based on the bound for real-valued random variables.

**Subgaussian confidence regime and convergence rates.** We now give an excess risk bound which is similar to the setting with bounded or subexponential noise: it exhibits a logarithmic dependence of the confidence parameter $\delta$ and a dependence of the sample size up to $n^{-1/3}$ depending on the level of smoothness given by $\nu$ [4, 6]. In the asymptotic large sample context, this bound is dominant and allows us to recover convergence rates.

**Corollary 3.2** (Subgaussian confidence regime). *Let* (MOM) *and* (SRC) *be satisfied. Then there exist constants* $\tilde{c}_1, \tilde{c}_2 > 0$ *such that with the regularization schedule*

$$\alpha_1(n,\delta) := \tilde{c}_2 \left( \frac{\log(6c_1/\delta)}{n} \right)^{\frac{1}{2\min\{\nu,1\}+1}}$$

*we have*

$$\|I_\pi \widehat{f}_{\alpha_1(n,\delta)} - f_\star\|_{L^2(\pi)} \leq \tilde{c}_1 R \left( \frac{\log(6c_1/\delta)}{n} \right)^{\frac{\min\{\nu,1\}}{2\min\{\nu,1\}+1}}, \tag{13}$$

*with confidence* $1 - \delta$ *for all* $\delta \in D_1(n,q)$ *and* $n \in \mathbb{N}$ *such that* $\alpha_1(n,\delta) \leq \kappa^2$.

The constants $\tilde{c}_1$ and $\tilde{c}_2$ in Corollary 3.2 only depend on $R, \nu, \kappa, \sigma, c_1, c_2$ and can be made explicit, but we omit their closed form here for the sake of a more accessible presentation. We refer the reader to the proof for more details.

**Remark 3.3** (Convergence rates). We directly obtain convergence rates from the above consideration. For *fixed confidence parameter* $\delta \in (0,1)$, we see that for all $n \geq n_0$, where $n_0$ is the effective sample size given in (12), we have $\delta \in D_1(n,q)$. Furthermore, there exists some $\tilde{n}_0 \in \mathbb{N}$ such that $\alpha_1(n,\delta) \leq \kappa^2$ for all $n \geq \tilde{n}_0$. Combining these two insights, from Corollary 3.2, we obtain

$$\|I_\pi \widehat{f}_{\alpha_1(n,\delta)} - f_\star\|_{L^2(\pi)} \leq \tilde{c}_1 R \left( \frac{\log(6c_1/\delta)}{n} \right)^{\frac{\min\{\nu,1\}}{2\min\{\nu,1\}+1}}$$

with confidence $1 - \delta$ for all $n \geq \max\{n_0, \tilde{n}_0\}$. We explicitly note that the convergence rates as well as the regularization schedule $\alpha_1(n,\delta)$ match exactly the known results for the capacity-independent setting that have been derived under the assumption of bounded or subexponential noise [4, 6].

**Polynomial confidence regime.** By definition (10), the polynomial confidence regime $\delta \in D_2(n,q)$ is relevant in the nonasymptotic investigation whenever $n < n_0$. For completeness, we address this setting in Appendix C and show that Proposition 3.1 can yield simplified risk bounds with suitable regularization schedules for $\alpha$ based on $\delta$ and $n$. Depending on $\delta$, these bounds may require a stronger regularization $\alpha_2(n,\delta)$ than the subgaussian confidence setting. In fact, the resulting bound exhibits a polynomial worst-case dependence on $\delta$, which is compensated by a better dependence on the sample size before transitioning to the behavior from the subgaussian confidence regime given by Corollary 3.2. This behavior can be observed in practice, which we confirm in a basic numerical experiment provided in Appendix A.

## 4 Capacity dependent bound and optimal rates

We now improve the results from the previous section and give an excess risk bound that involves the *effective dimension*, which has been established as a central tool to quantify the algorithm-dependent capacity of the hypothesis space $\mathcal{H}$ relative to the distribution $\pi$ and regularization parameter $\alpha$ in order to derive risk bounds in regularized kernel-based learning under the assumptions of an eigenvalue decay of $\mathcal{C}_\pi$ [3, 9, 10, 32].

**Definition 4.1** (Effective dimension). For $\alpha > 0$, we define

$$\mathcal{N}(\alpha) := \text{tr}\big(\mathcal{C}_\pi(\mathcal{C}_\pi + \alpha \,\text{Id}_{\mathcal{H}})^{-1}\big) < \infty.$$

We now assume the standard polynomial eigenvalue decay of $\mathcal{C}_\pi$ [3, 9, 11].

**Assumption 4.2** (Eigenvalue decay). We assume that the nonincreasingly ordered sequence of nonzero eigenvalues $(\mu_i)_{i \geq 1}$ of $\mathcal{T}_\pi$ satisfies the decay

$$\mu_i \leq D i^{-1/p}, \quad i \in \mathbb{N} \tag{EVD}$$

for a constant $D > 0$ and some $p \in (0,1)$.

Under the additional assumption (EVD), we can sharpen Proposition 3.1.

**Proposition 4.3** (Capacity-dependent excess risk bound). *Let* (MOM)*,* (SRC) *and* (EVD) *be satisfied. Suppose that* $\delta \in (0,1)$ *and*

$$\log(2/\delta)\left(\frac{2\kappa^2}{n\alpha} + \frac{2\sqrt{\tilde{D}}\kappa}{\sqrt{n}\alpha^{(1+p)/2}}\right) \leq 1. \tag{14}$$

*Then there exists a constant* $c > 0$ *not depending on* $\delta$ *and* $n$ *such that with confidence* $1 - \delta$*, we have*

$$\|I_\pi \widehat{f}_\alpha - f_\star\|_{L^2(\pi)} \leq c\left(\alpha^{\min\{\nu,1\}} + \frac{\log(8/\delta)}{\sqrt{\alpha}n} + \sqrt{\frac{\mathcal{N}(\alpha)\log(8/\delta)}{n}} + \sqrt{\mathcal{N}(\alpha)} \cdot \eta(\delta, n, \alpha)\right),$$

*where we set*

$$\eta(\delta, n, \alpha) := \max\left\{\left(\frac{1}{\delta n^{q-1}}\right)^{1/q} \cdot \left(\frac{1}{\alpha\mathcal{N}(\alpha)}\right)^{\frac{q-2}{2q}}, \sqrt{\frac{\log(8c_1/\delta)}{n}}\right\}.$$

The constant $c$ is made explicit in the proof. The key idea builds upon the original work of [3]. In particular, we incorporate the effective dimension into the Fuk–Nagaev inequality as a proxy for the $q$-th absolute moment appearing in the term $\eta(\delta, n, \alpha)$, thereby generalizing the idea to use the effective dimension as a variance proxy in the Bernstein inequality.

**Corollary 4.4** (Convergence rates). *Let* (MOM)*,* (SRC) *and* (EVD) *be satisfied. Then for every* $\delta \in (0,1)$*, there exists some* $n_0 \in \mathbb{N}$ *such that with the regularization schedule*

$$\alpha(n,\delta) := \left(\frac{\log(8c_1/\delta)}{n}\right)^{\frac{1}{2\min\{\nu,1\}+p}},$$

*we have*

$$\|I_\pi \widehat{f}_{\alpha(n,\delta)} - f_\star\|_{L^2(\pi)} \leq c\left(\frac{\log(8c_1/\delta)}{n}\right)^{\frac{\min\{\nu,1\}}{2\min\{\nu,1\}+p}} \tag{15}$$

*with confidence* $1 - \delta$ *for all* $n \geq n_0$ *with a constant* $c > 0$ *independent of* $n$ *and* $\delta$*.*

**Remark 4.5** (Optimality of rates). The rates provided by Corollary 4.4 match the rates from the literature derived for well-specified kernel ridge regression under the assumption of subexponential noise [3, 35]. In particular, these rates are known to be minimax optimal over the class of distributions satisfying (SRC), (EVD) and the Bernstein condition (3). Corollary 4.4 now proves that one can significantly relax the assumption of subexponential noise (3), as rate optimality is already achieved under the condition (MOM). Furthermore, the regularization schedule $\alpha(n,\delta)$ is the same as in the light-tailed setting—in particular, it does not depend on $q$.

## 5 Fuk–Nagaev inequality in Hilbert spaces

We discuss the central ingredient for the derivation of the previous results in more detail for convenience. We refer the reader to [24, 25] for the original work in the setting of real-valued random variables and [40] for a discussion of the involved constants. We present a sharpened version of a result due to [27, Theorem 3.5.1], which is formulated more generally in normed spaces, but exhibits an excess term that can be removed in the Hilbert space case. We provide the proof in Appendix D. We also note that the proof of the result in [27] is incomplete due to an inconsistent exponential moment bound. We address this issue by deriving an alternative bound.

**Proposition 5.1** (Fuk–Nagaev inequality; Hilbert space version). *Let* $\xi, \xi_1, \ldots \xi_n$ *be independent and identically distributed random variables taking values in a separable Hilbert space* $\mathcal{X}$ *such that*

$$\mathbb{E}[\xi] = 0, \quad \mathbb{E}[\|\xi\|_\mathcal{X}^2] < \sigma^2 \quad \text{and} \quad \mathbb{E}[\|\xi\|_\mathcal{X}^q] < Q,$$

*for some constants* $\sigma^2 > 0$*,* $Q > 0$ *and* $q \in \mathbb{N}$*,* $q \geq 3$*. Write* $S_n := \sum_{i=1} \xi_i$*. Then there exist two constants* $c_1 > 0$ *and* $c_2 > 0$ *depending only on* $q$ *such that for every* $t > 0$*, we have*

$$\mathbb{P}\left[\left\|\frac{1}{n}S_n\right\|_\mathcal{X} > t\right] \leq c_1\left(\frac{Q}{t^q n^{q-1}} + \exp\left(-c_2\frac{t^2 n}{\sigma^2}\right)\right). \tag{16}$$

**Remark 5.2.** For simplicity, we may assume that $1 \leq c_1$ when we apply Proposition 5.1.

**Confidence regimes.** Directly rearranging (16) from a tail bound to a confidence interval bound requires to solve a transcendental equation which does not admit a simple closed form solution. However, we can still derive an upper bound on the confidence intervals that reflects the superposition of polynomial and sub-gaussian tail in (16). By introducing

$$\delta := 2 \max\left\{ \frac{c_1 Q}{t^q n^{q-1}}, c_1 \exp\left(-c_2 \frac{t^2 n}{\sigma^2}\right) \right\}, \tag{17}$$

we have $\mathbb{P}[n^{-1}\|S_n\|_{\mathcal{X}} \leq t] \geq 1 - \delta$ by (16). Rearranging (17), we have

$$t \geq \left(\frac{2c_1 Q}{\delta n^{q-1}}\right)^{1/q} \quad \text{and} \quad t \geq \sigma \sqrt{\frac{\log(2c_1/\delta)}{c_2 n}}, \tag{18}$$

immediately leading to the following confidence bound.

**Corollary 5.3** (Confidence bound). *Under the assumptions of Proposition 5.1, for all $\delta \in (0,1)$, we have*

$$\left\|\frac{1}{n}S_n\right\|_{\mathcal{X}} \leq \max\left\{ \left(\frac{2c_1 Q}{\delta n^{q-1}}\right)^{1/q}, \sigma \sqrt{\frac{\log(2c_1/\delta)}{c_2 n}} \right\} \tag{19}$$

*with probability at least $1 - \delta$.*

For every fixed $\delta$ and $n \to \infty$, this shows the typical subgaussian behavior and a convergence rate of $\frac{1}{n}S_n$ of the order $n^{-1/2}$ with high probability. Interpreting the right hand side as a function of $\delta$ for a fixed sample size $n$ however, the above bound characterizes a *confidence regime change* at $\bar{\delta}(n)$, which we define as the solution to the equation

$$\left(\frac{2c_1 Q}{\bar{\delta}(n)\, n^{q-1}}\right)^{1/q} = \sigma \sqrt{\frac{\log(2c_1/\bar{\delta}(n))}{c_2 n}}. \tag{20}$$

In fact, in the *polynomial confidence regime* $\delta < \bar{\delta}(n)$, the dependence of the upper bound given in Corollary 5.3 on $\delta$ is clearly worse than in the *subgaussian regime* $\delta \geq \bar{\delta}(n)$ which is characterized by a logarithmic dependence on $\delta$. In contrast, the polynomial confidence regime allows for a better sample dependence of $n^{-(q-1)/q}$.

**Sharpness of the tail bound.** Both the subgaussian term and the polynomial term in the right hand side of the bound given by Proposition 5.1 can generally not be improved without additional assumptions. We repeat a similar argument as the one given in [63, Proposition 9], which is given in the context of linear processes. Let $\xi, \xi_1, \ldots, \xi_n$ be independent real-valued random variables drawn from a centered $t$-distribution with $q$ degrees of freedom, i.e. $\mathbb{E}[\xi^{q-c}] < \infty$ for all $0 < c \leq q$ and let $\sigma^2 := \mathbb{E}[\xi^2] = q/(q-2)$. Then [25, Theorem 1.9] shows that we have

$$\mathbb{P}[S_n/n > t] = \mathbb{P}[S_n/\sigma > nt/\sigma] \sim 1 - \phi(n^{1/2}t/\sigma) + n(1 - F_{\sigma^{-1}\xi}(nt/\sigma)) \quad \text{as } n \to \infty$$

for $nt/\sigma \geq n^{1/2}$, where $\Phi$ is the standard normal cumulative distribution function and $F_{\sigma^{-1}\xi}$ is the cumulative distribution function of $\sigma^{-1}\xi$. We now note that we have the basic property $F_{\sigma^{-1}\xi}(nt/\sigma) = F_\xi(nt)$ and we can show that the distribution of $\xi$ satisfies $1 - F(nt) \sim C_q/(nt)^q$ as $nt \to \infty$, where $C_q$ is a constant depending exclusively on $q$. In total, we obtain

$$\mathbb{P}[S_n/n > t] \sim 1 - \Phi(n^{1/2}t/\sigma) + \frac{C_q}{t^q n^{q-1}} \quad \text{as } n \to \infty$$

for $nt/\sigma \geq n^{1/2}$, showing that Proposition 5.1 is asymptotically optimal.

## Acknowledgements

Dimitri Meunier and Arthur Gretton are supported by the Gatsby Charitable Foundation. This work was partially funded by the German Federal Ministry of Research, Technology and Space as part of the 6G-CAMPUS project (grant no. 16KISK194). The authors express their gratitude to Christian Fiedler for pointing out inconsistensies in the proof of [27, Lemma 3.5.1] and for his contribution to the corresponding corrections in Appendix D.3.1 and Appendix D.3.3.

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

# Appendices

The appendices are organized as follows.

Appendix A contains a numerical experiment which confirms the behavior of the excess risk as described by the bounds provided in the main text in a very basic setting. In Appendix B, we report proofs of the results in the main text of this paper. In Appendix C, we provide an additional excess risk bound for the polynomial confidence regime based on Proposition 3.1. We collect some tail bounds and concentration results in Appendix D. In particular, Appendix D.3 contains a proof of Proposition 5.1. In Appendix E, we recall additional miscellaneous inequalities required for our proofs.

## A  Numerical experiment

We provide a basic numerical experiment[3] showing that the confidence regimes are not only occurring in the upper bound on the excess risk given by Proposition 3.1, but are effectively reflected in the distribution of the excess risk in practice, even for very simple models.

We consider the input space $\mathcal{X} := \mathbb{R}$ equipped with the RKHS $\mathcal{H}$ induced by the radial basis kernel $k(x_1, x_2) := \exp(\frac{-|x_1 - x_2|^2}{2})$ and define the target function $f_\star(x) := \sum_{i=1}^5 a_i k(x_i, x) \in \mathcal{H}$, $x \in \mathcal{X}$ for vectors $\mathbf{a} := (2, -1, -3, 1, 2)$ and $\mathbf{x} := (-4, -2 - 0, 3, 7)$. We define the covariate distribution $X \sim \pi := \mathcal{N}(0, 1)$ on $\mathcal{X}$ and generate independent observation pairs with a light-tailed noise distribution and a heavy-tailed distribution with identical variance based on

    (i)  the *light-tailed noise model* given by

$$Y_{(\mathcal{N})} = f_\star(X) + \varepsilon_{(\mathcal{N})}, \tag{21}$$

        where the noise $\varepsilon_{(\mathcal{N})} \sim \mathcal{N}(0, \sigma^2)$ follows a centered Gaussian distribution with variance $\sigma^2 = 3$.

---

[3]We provide the source code on GitHub: `https://github.com/mollenhauerm/krr-heavy-tailed`. The experiment can be run on the CPU of any consumer laptop.

(ii) the *heavy-tailed noise model* with a finite number of higher moments given by

$$Y_{(t)} = f_\star(X) + \varepsilon_{(t)}, \tag{22}$$

where the noise $\varepsilon_{(t)} \sim \mathbf{t}(0, \nu)$ follows a centered $t$-distribution with $\nu = 3$ degrees of freedom and $\mathbb{E}[\varepsilon_{(t)}^2] = 3$.

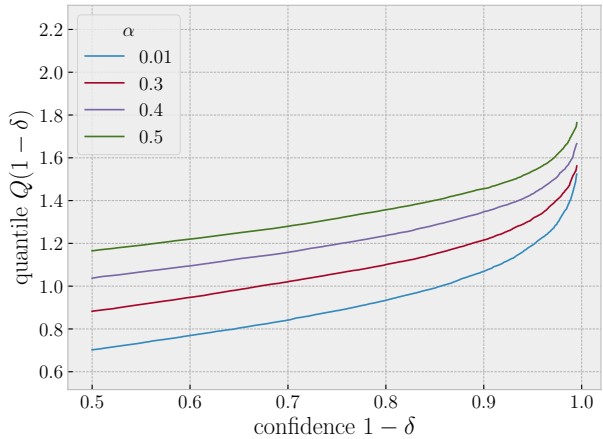

(a) Light-tailed noise

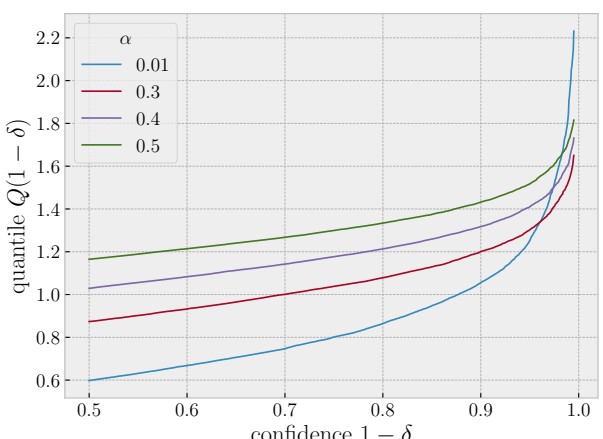

(b) Heavy-tailed noise

Figure 2: Empirical approximations of the quantile function of the excess risk $\|I_\pi \widehat{f}_\alpha - f_\star\|_{L^2(\pi)}$ for (a) the light-tailed noise model given in (21) and (b) in the $t$-distributed noise model given by (22) for different choices of the regularization parameter $\alpha$ and fixed sample size $n = 20$.

For both models, we compute $\widehat{f}_\alpha$ based on the generated sample pairs for a sample size $n = 20$ and record the error $\|I_\pi \widehat{f}_\alpha - f_\star\|_{L^2(\pi)}$, which we approximate through Monte Carlo simulation by drawing samples from $\pi$. We perform the above computation for a selection of different regularization parameters $\alpha$, repeating each experiment across 10000 random seeds (per model and choice of $\alpha$). Based on the recorded errors, we empirically approximate the quantile function of the distributions of $\|I_\pi \widehat{f}_\alpha - f_\star\|_{L^2(\pi)}$ given by

$$Q(1-\delta) := \inf\left\{ t : \mathbb{P}\left[\|I_\pi \widehat{f}_\alpha - f_\star\|_{L^2(\pi)} \le t\right] \ge 1 - \delta \right\}, \quad \delta \in [0,1]$$

The resulting approximated quantile functions are visualized in Figure 2 for all choices of $\alpha$ and both models.

While the excess risk quantiles exhibit the same qualitative behaviour and are of the same order in both models for moderate confidence levels, the excess risk quantiles of the heavy-tailed model increase rapidly for a small regularization parameter beyond a high confidence level of $1 - \delta \approx 0.95$. Consequently, the heavy-tailed model requires a stronger regularization than the light-tailed model in high confidence settings, as explained by the transition of the excess risk from a subgaussian confidence regime to a polynomial confidence regime in Section 3.

## B  Proofs

We provide the proofs for the results presented in the main text of this work.

### B.1  Proof of Proposition 3.1

We investigate the classical *bias-variance decomposition*

$$I_\pi \widehat{f}_\alpha - f_\star = I_\pi \widehat{f}_\alpha - I_\pi f_\alpha + I_\pi f_\alpha - f_\star$$

and bound both terms individually.

#### B.1.1  Bounding the variance $I_\pi \widehat{f}_\alpha - I_\pi f_\alpha$.

We first collect some generic properties of $f_\alpha$. The identity $f_\alpha = (\mathcal{C}_\pi + \alpha \operatorname{Id}_{\mathcal{H}})^{-1} I_\pi^* f_\star$ gives

$$\mathbb{E}[\widehat{I}_\pi^*(f_\star - \widehat{I}_\pi f_\alpha)] = I_\pi^*(f_\star - I_\pi f_\alpha) = \alpha f_\alpha. \tag{23}$$

Based on (6), (7) and (8), we decompose the variance in $\mathcal{H}$ as

$$
\begin{aligned}
\widehat{f}_\alpha - f_\alpha &= (\widehat{\mathcal{C}}_\pi + \alpha \operatorname{Id}_{\mathcal{H}})^{-1} \left\{ \widehat{I}_\pi^*(f_\star - \widehat{I}_\pi f_\alpha) - \alpha f_\alpha + \frac{1}{n}\sum_{i=1}^n \phi(X_i)\varepsilon_i \right\} \\
&= (\widehat{\mathcal{C}}_\pi + \alpha \operatorname{Id}_{\mathcal{H}})^{-1} \left\{ \widehat{I}_\pi^*(f_\star - \widehat{I}_\pi f_\alpha) - I_\pi^*(f_\star - I_\pi f_\alpha) + \frac{1}{n}\sum_{i=1}^n \phi(X_i)\varepsilon_i \right\} \quad \text{by (23)} \\
&= (\widehat{\mathcal{C}}_\pi + \alpha \operatorname{Id}_{\mathcal{H}})^{-1}\{\Delta_1 + \Delta_2\},
\end{aligned}
$$

where we introduce the $\mathcal{H}$-valued random variables

$$\Delta_1 = \widehat{I}_\pi^*(f_\star - \widehat{I}_\pi f_\alpha) - \mathbb{E}[\widehat{I}_\pi^*(f_\star - \widehat{I}_\pi f_\alpha)], \qquad \Delta_2 = \frac{1}{n}\sum_{i=1}^n \phi(X_i)\varepsilon_i . \tag{24}$$

We now obtain an $L^2(\pi)$-norm bound on the variance as

$$
\begin{aligned}
\|I_\pi(\widehat{f}_\alpha - f_\alpha)\|_{L^2(\pi)} = \|\mathcal{C}_\pi^{1/2}(\widehat{f}_\alpha - f_\alpha)\|_{\mathcal{H}} &= \|\mathcal{C}_\pi^{1/2}(\widehat{\mathcal{C}}_\pi + \alpha \operatorname{Id}_{\mathcal{H}})^{-1}\{\Delta_1 + \Delta_2\}\|_{\mathcal{H}} \\
&\leq \sqrt{2}\, \mathcal{B}(n,\delta,\alpha)^{1/2} \left\| (\widehat{\mathcal{C}}_\pi + \alpha \operatorname{Id}_{\mathcal{H}})^{1/2}(\widehat{\mathcal{C}}_\pi + \alpha \operatorname{Id}_{\mathcal{H}})^{-1}\{\Delta_1 + \Delta_2\} \right\|_{\mathcal{H}} \\
&\leq 2\sqrt{2} \left\| (\widehat{\mathcal{C}}_\pi + \alpha \operatorname{Id}_{\mathcal{H}})^{-1/2}\{\Delta_1 + \Delta_2\} \right\|_{\mathcal{H}} \\
&\leq 2\sqrt{\frac{2}{\alpha}}(\|\Delta_1\|_{\mathcal{H}} + \|\Delta_2\|_{\mathcal{H}}), \tag{25}
\end{aligned}
$$

with probability at least $1 - \delta$, where $\mathcal{B}(n,\delta,\alpha)$ is defined in (57) and where we apply Lemma D.10 in the first inequality, Corollary D.7 in the second inequality, and additionally the fact that we have $\|(\widehat{\mathcal{C}}_\pi + \alpha \operatorname{Id}_{\mathcal{H}})^{-1/2}\|_{\mathcal{L}(\mathcal{H})} \leq \alpha^{-1/2}$ in the last inequality.

We now provide individual confidence bounds for $\Delta_1$ and $\Delta_2$.

**Bounding $\Delta_1$: Bennett inequality.**  We introduce the independent and identically distributed random variables $\xi_i := (f_\star(X_i) - f_\alpha(X_i))\phi(X_i) \in \mathcal{H}$ and note that $\Delta_1 := \frac{1}{n}\sum_{i=1}^n (\xi_i - \mathbb{E}[\xi_i])$ and proceed similarly as in the proof of [4, Theorem 1 & Theorem 5].

To apply Bennett's inequality, we need to bound the norm of the $\xi_i$. First, note that almost surely, we have by (SRC), for some $f \in L^2(\pi)$, such that $\|f\|_{L^2(\pi)} \leq R$

$$
\begin{aligned}
\|f_\alpha(X_i)\phi(X_i)\|_\mathcal{H} &= \|\langle f_\alpha, \phi(X_i)\rangle_\mathcal{H}\phi(X_i)\|_\mathcal{H} \\
&\leq \kappa^2 \|f_\alpha\|_\mathcal{H} \\
&= \kappa^2 \|(\mathcal{C}_\pi + \alpha\,\mathrm{Id}_\mathcal{H})^{-1} I_\pi^* f_\star\|_\mathcal{H} \\
&= \kappa^2 \|I_\pi^*(\mathcal{T}_\pi + \alpha\,\mathrm{Id}_\mathcal{H})^{-1}\mathcal{T}_\pi^\nu f\|_\mathcal{H}.
\end{aligned}
$$

A short calculation shows that for $\nu \geq 1/2$, the map $h(t) = t^{\nu+1/2}(t+\alpha)^{-1}$ is increasing on the spectrum of $\mathcal{T}_\pi$ with

$$
\sup_{t\in(0,\kappa^2]} |h(t)| = \sup_{t\in(0,\kappa^2]} t^{\nu-1/2}\frac{t}{t+\alpha} \leq \kappa^{2(\nu-1/2)},
$$

for all $\alpha > 0$. Hence,

$$
\|I_\pi^*(\mathcal{T}_\pi + \alpha\,\mathrm{Id}_\mathcal{H})^{-1}\mathcal{T}_\pi^\nu f\|_\mathcal{H} \leq R \cdot \sup_{t\in(0,\kappa^2]} |h(t)| \leq R \cdot \kappa^{2(\nu-1/2)}
$$

and therefore

$$
\|f_\alpha(X_i)\phi(X_i)\|_\mathcal{H} \leq R \cdot \kappa^{2\nu+1} .
$$

This gives

$$
\begin{aligned}
\|\xi_i\|_\mathcal{H} &\leq \kappa\|f_\star\|_{L^\infty(\pi)} + \|f_\alpha(X_i)\phi(X_i)\|_\mathcal{H} \\
&\leq \kappa\|f_\star\|_{L^\infty(\pi)} + R \cdot \kappa^{2\nu+1}.
\end{aligned}
$$

Furthermore, by [64, Proposition 3.3] and [11, Lemma 25] we see

$$
\begin{aligned}
\mathbb{E}[\|\xi_i\|_\mathcal{H}^2] &\leq \kappa^2\|f_\star - I_\pi f_\alpha\|_{L^2(\pi)}^2 \\
&= \kappa^2\|(\mathrm{Id}_\mathcal{H} - \mathcal{T}_\pi(\mathcal{T}_\pi + \alpha\,\mathrm{Id}_\mathcal{H})^{-1})\mathcal{T}_\pi^\nu f\|_{L^2(\pi)}^2 \\
&\leq \kappa^2 R^2 \alpha^{2\min\{\nu,1\}}.
\end{aligned}
$$

Applying Proposition D.2 to $\Delta_1 = \frac{1}{n}\sum_{i=1}^n (\xi_i - \mathbb{E}[\xi_i])$, with probability at least $1-\delta$ for for any $1 > \delta > 0$, we have

$$
\begin{aligned}
\|\Delta_1\|_\mathcal{H} &\leq \frac{2(\kappa\|f_\star\|_{L^\infty(\pi)} + R\cdot\kappa^{2\nu+1})\log(2/\delta)}{n} + \sqrt{\frac{2\kappa^2 R^2 \alpha^{2\min\{\nu,1\}}\log(2/\delta)}{n}} \\
&\leq \tilde{C}_\diamond \cdot \left(\frac{\log(2/\delta)}{n} + \sqrt{\frac{\alpha^{2\min\{\nu,1\}}\log(2/\delta)}{n}}\right),
\end{aligned}
\tag{26}
$$

where we use $\sqrt{2\log(2/\delta)} \leq 2\log(2/\delta)$ and with

$$
\tilde{C}_\diamond = 2\kappa \cdot \max\{\|f_\star\|_{L^\infty(\pi)} + R\cdot\kappa^{2\nu}, R\}.
\tag{27}
$$

**Bounding $\Delta_2$: Fuk–Nagaev inequality.**   We introduce the independent and identically distributed random variables $\zeta_i := \phi(X_i)\varepsilon_i$ and note that $\Delta_2 = \frac{1}{n}\sum_{i=1}^n \zeta_i$. By Assumption 2.2, we clearly have $\mathbb{E}[\zeta_i] = \mathbb{E}[\phi(X_i)\mathbb{E}[\varepsilon_i \mid X]] = 0$ as well as

$$
\mathbb{E}[\|\zeta_i\|_\mathcal{H}^2] < \kappa^2\sigma^2 \quad \text{and} \quad \mathbb{E}[\|\zeta_i\|_\mathcal{H}^q] < \kappa^q Q.
$$

We can therefore apply Corollary 5.3 to $\Delta_2 = \frac{1}{n}\sum_{i=1}^n \zeta_i$ and obtain

$$
\|\Delta_2\|_\mathcal{H} \leq \max\left\{\left(\frac{2c_1\kappa^q Q}{\delta n^{q-1}}\right)^{1/q}, \kappa\sigma\sqrt{\frac{\log(2c_1/\delta)}{c_2 n}}\right\}
\tag{28}
$$

with probability at least $1-\delta$.

**Collecting the terms.** By the union bound, the three individual confidence bounds (25), (26) and (28) and hold simultaneously with probability at least $1 - \delta$. We have

$$\|I_\pi(\widehat{f}_\alpha - f_\alpha)\|_{L^2(\pi)} \leq \frac{C_\diamond}{\sqrt{\alpha}}\left(\frac{\log(6/\delta)}{n} + \sqrt{\frac{\alpha^{2\min\{\nu,1\}}\log(6/\delta)}{n}} + \eta(\delta, n)\right).$$

Here, we set

$$\eta(\delta, n) := \max\left\{\left(\frac{Q}{\delta n^{q-1}}\right)^{1/q}, \sigma\sqrt{\frac{\log(6c_1/\delta)}{n}}\right\}$$

and

$$C_\diamond = 2\sqrt{2}\max\{\tilde{C}_\diamond, \kappa \cdot \max\{(6c_1)^{1/q}, 1/\sqrt{c_2}\}\}. \tag{29}$$

### B.1.2  Bounding the bias $I_\pi f_\alpha - f_\star$

We apply the standard theory of source conditions in inverse problems as discussed for example by [59]. In particular, under the $L^2(\pi)$-source condition given by (SRC), we have

$$\|I_\pi f_\alpha - f_\star\|_{L^2(\pi)} \leq R\alpha^{\min\{\nu,1\}},$$

see e.g. [64, Proposition 3.3]. Combining the variance bound and the bias bound yields the result.

### B.2  Proof of Corollary 3.2

Let $\delta \in D_1(n, q)$. Using that $\alpha \leq \kappa^2$ and recalling that $c_1 \geq 1$, we find by Proposition 3.1 for any $\delta \in D_1(n, q)$ with confidence $1 - \delta$

$$\|I_\pi(\widehat{f}_\alpha - f_\star)\|_{L^2(\pi)} \leq R\alpha^{\min\{\nu,1\}} + \frac{C_\diamond}{\sqrt{\alpha}}\left(\frac{\log(6/\delta)}{n} + \sqrt{\frac{\alpha^{2\min\{\nu,1\}}\log(6/\delta)}{n}} + \sigma\sqrt{\frac{\log(6c_1/\delta)}{n}}\right).$$

Assuming

$$\alpha^{\min\{\nu,1\}}\sqrt{n} \geq \sqrt{\log(6/\delta)} \tag{30}$$

gives

$$\frac{\log(6/\delta)}{n} \leq \sqrt{\frac{\alpha^{2\min\{\nu,1\}}\log(6/\delta)}{n}} \leq \kappa^{2\min\{\nu,1\}}\sqrt{\frac{\log(6c_1/\delta)}{n}}.$$

Hence,

$$\|I_\pi\widehat{f}_\alpha - f_\star\|_{L^2(\pi)} \leq R\alpha^{\min\{\nu,1\}} + c_3\sqrt{\frac{\log(6c_1/\delta)}{\alpha n}},$$

with $c_3 = 3C_\diamond\max\{\kappa^{2\min\{\nu,1\}}, \sigma\}$. Ensuring that the remaining estimation error contribution will not be dominated by the approximation error, we see that the regularization parameter has to satisfy

$$\alpha \geq c_4\left(\frac{\log(6c_1/\delta)}{n}\right)^{\frac{1}{2\min\{\nu,1\}+1}}, \tag{31}$$

with $c_4 = (c_3/R)^{\frac{1}{\min\{\nu,1\}+1/2}}$. We therefore obtain

$$\|I_\pi\widehat{f}_\alpha - f_\star\|_{L^2(\pi)} \leq \tilde{c}_1 R\left(\frac{\log(6c_1/\delta)}{n}\right)^{\frac{\min\{\nu,1\}}{2\min\{\nu,1\}+1}},$$

with confidence $1 - \delta$ and $\tilde{c}_1 = 2c_4^{\min\{\nu,1\}}$. In order for this bound to hold, the regularization strength has to fulfill condition (9), (30) and (31), that is,

$$\alpha \geq \max\left\{c_4\left(\frac{\log(6c_1/\delta)}{n}\right)^{\frac{1}{2\min\{\nu,1\}+1}}, \left(\frac{\log(6/\delta)}{n}\right)^{\frac{1}{2\min\{\nu,1\}}}, C_\kappa\frac{\log(6/\delta)}{\sqrt{n}}\right\} \tag{32}$$

in addition to the restriction $\alpha \leq \kappa^2$. As we always have $n^{-\frac{1}{2\min\{\nu,1\}}} \leq n^{-1/2} \leq n^{-\frac{1}{2\min\{\nu,1\}+1}}$ for $\nu \geq 1/2$ and $\log(6c_1/\delta) \geq \log(6/\delta)$, there exists a constant $\tilde{c}_2$ such that

$$\alpha_1(n, \delta) := \tilde{c}_2\left(\frac{\log(6c_1/\delta)}{n}\right)^{\frac{1}{2\min\{\nu,1\}+1}}$$

satisfies (32).

### B.3  Proof of Proposition 4.3

We split again into *variance* and *bias*:

$$I_\pi \widehat{f}_\alpha - f_\star = I_\pi(\widehat{f}_\alpha - f_\alpha) + I_\pi f_\alpha - f_\star.$$

#### B.3.1  Bounding the variance $I_\pi \widehat{f}_\alpha - I_\pi f_\alpha$

Following [65, Appendix C], we further decompose the variance term in $L^2(\pi)$ as

$$
\begin{aligned}
I_\pi \widehat{f}_\alpha - I_\pi f_\alpha &= I_\pi \widehat{f}_\alpha - I_\pi \widehat{\mathcal{C}}_\pi (\widehat{\mathcal{C}}_\pi + \alpha \operatorname{Id}_{\mathcal{H}})^{-1} f_\alpha + I_\pi \widehat{\mathcal{C}}_\pi (\widehat{\mathcal{C}}_\pi + \alpha \operatorname{Id}_{\mathcal{H}})^{-1} f_\alpha - I_\pi f_\alpha \\
&= \underbrace{I_\pi (\widehat{\mathcal{C}}_\pi + \alpha \operatorname{Id}_{\mathcal{H}})^{-1}\left(\widehat{\Upsilon} - \widehat{\mathcal{C}}_\pi f_\alpha\right)}_{=:I_1} + \underbrace{\alpha I_\pi (\widehat{\mathcal{C}}_\pi + \alpha \operatorname{Id}_{\mathcal{H}})^{-1} f_\alpha}_{=:I_2},
\end{aligned}
\tag{33}
$$

where we use the definition of $\widehat{f}_\alpha$ in (7) for the first term and the identity

$$\widehat{\mathcal{C}}_\pi (\widehat{\mathcal{C}}_\pi + \alpha \operatorname{Id}_{\mathcal{H}})^{-1} - \operatorname{Id}_{\mathcal{H}} = \alpha(\widehat{\mathcal{C}}_\pi + \alpha \operatorname{Id}_{\mathcal{H}})^{-1}$$

for the second summand.

#### B.3.2  Bounding the first term $I_1$

We introduce the shorthand notation $\mathcal{C}_\alpha := \mathcal{C}_\pi + \alpha \operatorname{Id}_{\mathcal{H}}, \widehat{\mathcal{C}}_\alpha := \widehat{\mathcal{C}}_\pi + \alpha \operatorname{Id}_{\mathcal{H}}$. We note that we may write

$$\mathcal{C}_\pi^{1/2}(\widehat{\mathcal{C}}_\pi + \alpha \operatorname{Id}_{\mathcal{H}})^{-1} = \mathcal{C}_\pi^{1/2}\mathcal{C}_\alpha^{-1/2}\mathcal{C}_\alpha^{1/2}\widehat{\mathcal{C}}_\alpha^{-1/2}\widehat{\mathcal{C}}_\alpha^{-1/2}\mathcal{C}_\alpha^{1/2}\mathcal{C}_\alpha^{-1/2}.$$

Recall that $\|\mathcal{C}_\pi^{1/2}\mathcal{C}_\alpha^{-1/2}\|_{\mathcal{L}(\mathcal{H})} \le 1$ and that by Corollary D.9 and Lemma E.2, with confidence $1 - \delta$, we have

$$\|\mathcal{C}_\alpha^{1/2}\widehat{\mathcal{C}}_\alpha^{-1/2}\|_{\mathcal{L}(\mathcal{H})} = \|\widehat{\mathcal{C}}_\alpha^{-1/2}\mathcal{C}_\alpha^{1/2}\|_{\mathcal{L}(\mathcal{H})} \le 2,$$

provided (14) is satisfied. Hence, by (4), with confidence $1 - \delta$

$$\|I_1\|_{L^2(\pi)} = \|I_\pi(\widehat{\mathcal{C}}_\pi + \alpha \operatorname{Id}_{\mathcal{H}})^{-1}(\widehat{\Upsilon} - \widehat{\mathcal{C}}_\pi f_\alpha)\|_{L^2(\pi)} \le 4\,\|(\mathcal{C}_\pi + \alpha \operatorname{Id}_{\mathcal{H}})^{-1/2}(\widehat{\Upsilon} - \widehat{\mathcal{C}}_\pi f_\alpha)\|_{\mathcal{H}}.$$
$$\tag{34}$$

We proceed by further splitting

$$
\begin{aligned}
(\mathcal{C}_\pi + \alpha \operatorname{Id}_{\mathcal{H}})^{-1/2}(\widehat{\Upsilon} - \widehat{\mathcal{C}}_\pi f_\alpha) &= (\mathcal{C}_\pi + \alpha \operatorname{Id}_{\mathcal{H}})^{-1/2}[(\widehat{\Upsilon} - \widehat{\mathcal{C}}_\pi f_\alpha) - (I_\pi^* f_\star - \mathcal{C}_\pi f_\alpha)] \\
&\quad + (\mathcal{C}_\pi + \alpha \operatorname{Id}_{\mathcal{H}})^{-1/2}(I_\pi^* f_\star - \mathcal{C}_\pi f_\alpha).
\end{aligned}
$$

Note that by (8), we have

$$\widehat{\Upsilon} = \widehat{I}_\pi^* f_\star + \frac{1}{n}\sum_{i=1}^n \phi(X_i)\varepsilon_i.$$

Introducing

$$\Delta_1 := (\widehat{I}_\pi^* f_\star - \widehat{\mathcal{C}}_\pi f_\alpha) - (I_\pi^* f_\star - \mathcal{C}_\pi f_\alpha), \qquad \Delta_2 := \frac{1}{n}\sum_{i=1}^n \phi(X_i)\varepsilon_i,$$

we obtain

$$(\mathcal{C}_\pi + \alpha \operatorname{Id}_{\mathcal{H}})^{-1/2}(\widehat{\Upsilon} - \widehat{\mathcal{C}}_\pi f_\alpha) = I_{11} + I_{12} + I_{13},$$

where we set

$$
\begin{aligned}
I_{11} &:= (\mathcal{C}_\pi + \alpha \operatorname{Id}_{\mathcal{H}})^{-1/2}\Delta_1 \\
I_{12} &:= (\mathcal{C}_\pi + \alpha \operatorname{Id}_{\mathcal{H}})^{-1/2}\Delta_2 \\
I_{13} &:= (\mathcal{C}_\pi + \alpha \operatorname{Id}_{\mathcal{H}})^{-1/2}(I_\pi^* f_\star - \mathcal{C}_\pi f_\alpha).
\end{aligned}
$$

Thus, with (34), we have

$$\|I_1\|_{L^2(\pi)} \le 4(\|I_{11}\|_{\mathcal{H}} + \|I_{12}\|_{\mathcal{H}} + \|I_{13}\|_{\mathcal{H}}) \tag{35}$$

We proceed to bound the terms on the right hand side individually.

**Bounding $\|I_{11}\|_{\mathcal{H}}$.** To bound the norm of $I_{11}$, we apply Bennett's inequality Proposition D.2 to the independent and identically distributed random variables

$$\xi_i := (\mathcal{C}_\pi + \alpha \operatorname{Id}_{\mathcal{H}})^{-1/2}(f_\star(X_i) - f_\alpha(X_i))\phi(X_i).$$

Then

$$I_{11} = \frac{1}{n} \sum_{i=1}^{n} \xi_i - \mathbb{E}[\xi_i].$$

Furthermore,

$$\mathbb{E}[\|\xi_i\|_{\mathcal{H}}^2] \leq \sup_{X \in \mathcal{X}} |f_\star(X) - f_\alpha(X)|^2 \, \mathbb{E}[\|(\mathcal{C}_\pi + \alpha \operatorname{Id}_{\mathcal{H}})^{-1/2}\phi(X_i)\|_{\mathcal{H}}^2].$$

since $f_\alpha \in \mathcal{H}$ and by (SRC), we have for $\nu \geq 1/2$

$$
\begin{aligned}
|f_\alpha(X)| &= |\langle f_\alpha, \phi(X)\rangle_{\mathcal{H}}| \\
&= |\langle (\mathcal{C}_\pi + \alpha \operatorname{Id}_{\mathcal{H}})^{-1} I_\pi^* \mathcal{T}_\pi^\nu f, \phi(X)\rangle_{\mathcal{H}}| \\
&= |\langle f, \mathcal{T}_\pi^\nu I_\pi (\mathcal{C}_\pi + \alpha \operatorname{Id}_{\mathcal{H}})^{-1}\phi(X)\rangle_{L^2(\pi)}| \\
&\leq \kappa R \|I_\pi^*(\mathcal{T}_\pi + \alpha \operatorname{Id}_{L^2(\pi)})^{-1}\mathcal{T}_\pi^\nu\|_{\mathcal{L}(L^2(\pi),\mathcal{H})} \\
&\leq \kappa R \sup_{0 < t \leq \kappa^2} |t^{\nu+1/2}(t+\alpha)^{-1}| \\
&\leq \kappa^{2\nu} R.
\end{aligned}
\tag{36}
$$

We proceed with writing

$$\mathbb{E}[\|(\mathcal{C}_\pi + \alpha \operatorname{Id}_{\mathcal{H}})^{-1/2}\phi(X_i)\|_{\mathcal{H}}^2] = \mathbb{E}[\operatorname{tr}((\mathcal{C}_\pi + \alpha \operatorname{Id}_{\mathcal{H}})^{-1}\phi(X_i) \otimes \phi(X_i))] = \mathcal{N}(\alpha).$$

Hence,

$$\mathbb{E}[\|\xi_i\|_{\mathcal{H}}^2] \leq 2(\|f_\star\|_\infty^2 + \kappa^{4\nu} R^2)\mathcal{N}(\alpha) =: \tilde{\sigma}^2.$$

Moreover,

$$
\begin{aligned}
\|\xi_i\|_{\mathcal{H}} &= \|(\mathcal{C}_\pi + \alpha \operatorname{Id}_{\mathcal{H}})^{-1/2}(f_\star(X_i) - f_\alpha(X_i))\phi(X_i)\|_{\mathcal{H}} \\
&\leq \|f_\star(X_i)(\mathcal{C}_\pi + \alpha \operatorname{Id}_{\mathcal{H}})^{-1/2}\phi(X_i)\|_{\mathcal{H}} + \|f_\alpha(X_i)(\mathcal{C}_\pi + \alpha \operatorname{Id}_{\mathcal{H}})^{-1/2}\phi(X_i)\|_{\mathcal{H}} \\
&\leq \frac{\kappa\|f_\star\|_\infty}{\sqrt{\alpha}} + \frac{\kappa f_\alpha(X_i)}{\sqrt{\alpha}} \\
&\leq \frac{\kappa}{\sqrt{\alpha}} (\|f_\star\|_\infty + \kappa^{2\nu} R) =: L.
\end{aligned}
$$

In the last step we use (36). From Proposition D.2, we obtain with confidence $1 - \delta$

$$
\begin{aligned}
\|I_{11}\|_{\mathcal{H}} &\leq \frac{2L \log(2/\delta)}{n} + \sqrt{\frac{2\tilde{\sigma}^2 \log(2/\delta)}{n}} \\
&\leq c_{11}\left(\frac{\log(2/\delta)}{\sqrt{\alpha}n} + \sqrt{\frac{\mathcal{N}(\alpha) \log(2/\delta)}{n}}\right),
\end{aligned}
\tag{37}
$$

with $c_{11} = 2(\|f_\star\|_\infty + \kappa^{2\nu} R) \max\{1, \kappa\}$.

**Bounding $\|I_{12}\|_{\mathcal{H}}$.** The idea is to apply the Fuk-Nagaev inequality Corollary 5.3 to the independent and identically distributed random variables

$$\zeta_i := \varepsilon_i (\mathcal{C}_\pi + \alpha \operatorname{Id}_{\mathcal{H}})^{-1/2}\phi(X_i).$$

We repeatedly apply Assumption 2.2 to bound the expectation and moments of the $\zeta_i$. We get

$$\mathbb{E}[\zeta_i] = \mathbb{E}[\underbrace{\mathbb{E}[\varepsilon_i|X_i]}_{=0}(\mathcal{C}_\pi + \alpha \operatorname{Id}_{\mathcal{H}})^{-1/2}\phi(X_i)] = 0.$$

Moreover,

$$\mathbb{E}[\|\zeta_i\|_{\mathcal{H}}^2] = \mathbb{E}[\mathbb{E}[\varepsilon_i^2|X_i]\,\|(\mathcal{C}_\pi + \alpha\,\mathrm{Id}_{\mathcal{H}})^{-1/2}\phi(X_i)\|_{\mathcal{H}}^2]$$
$$< \sigma^2 \mathbb{E}[\mathrm{tr}((\mathcal{C}_\pi + \alpha\,\mathrm{Id}_{\mathcal{H}})^{-1}\phi(X_i) \otimes \phi(X_i))]$$
$$= \sigma^2 \mathcal{N}(\alpha)$$
$$=: \tilde{\sigma}^2.$$

Since $q > 2$, we find

$$\mathbb{E}[\|\zeta_i\|_{\mathcal{H}}^q] = \mathbb{E}[\mathbb{E}[|\varepsilon_i|^q|X_i]\,\|(\mathcal{C}_\pi + \alpha\,\mathrm{Id}_{\mathcal{H}})^{-1/2}\phi(X_i)\|_{\mathcal{H}}^q]$$
$$< Q\,\mathbb{E}[\|(\mathcal{C}_\pi + \alpha\,\mathrm{Id}_{\mathcal{H}})^{-1/2}\phi(X_i)\|_{\mathcal{H}}^2\|(\mathcal{C}_\pi + \alpha\,\mathrm{Id}_{\mathcal{H}})^{-1/2}\phi(X_i)\|_{\mathcal{H}}^{q-2}]$$
$$\leq Q\,\kappa^{q-2}\left(\frac{1}{\alpha}\right)^{\frac{q-2}{2}}\mathcal{N}(\alpha)$$
$$=: \tilde{Q}.$$

Since $\frac{1}{n}\sum_{i=1}^n \zeta_i = I_{12}$, applying Corollary 5.3 gives with confidence $1 - \delta$

$$\|I_{12}\|_{\mathcal{H}} \leq \max\left\{\left(\frac{2c_1\tilde{Q}}{\delta n^{q-1}}\right)^{1/q}, \tilde{\sigma}\sqrt{\frac{\log(2c_1/\delta)}{c_2 n}}\right\}$$
$$= \max\left\{\left(\frac{2c_1\kappa^{q-2}Q}{\delta n^{q-1}}\right)^{1/q}\left(\frac{1}{\alpha}\right)^{\frac{q-2}{2q}}\mathcal{N}(\alpha)^{1/q}, \sigma\sqrt{\frac{\mathcal{N}(\alpha)\log(2c_1/\delta)}{c_2 n}}\right\}$$
$$\leq c_{12}\sqrt{\mathcal{N}(\alpha)}\,\max\left\{\left(\frac{1}{\delta n^{q-1}}\right)^{1/q}\left(\frac{1}{\alpha\mathcal{N}(\alpha)}\right)^{\frac{q-2}{2q}}, \sqrt{\frac{\log(2c_1/\delta)}{n}}\right\}, \tag{38}$$

with $c_{12} = \max\{(2c_1\kappa^{q-2}Q)^{1/q}, \sigma/\sqrt{c_2}\}$.

**Bounding $\|I_{13}\|_{\mathcal{H}}$.** Using the definition of $f_\alpha$ from (6), we see that by (SRC), we have

$$I_{13} = (\mathcal{C}_\pi + \alpha\,\mathrm{Id}_{\mathcal{H}})^{-1/2}(I_\pi^* f_\star - \mathcal{C}_\pi(\mathcal{C}_\pi + \alpha\,\mathrm{Id}_{\mathcal{H}})^{-1}I_\pi^* f_\star)$$
$$= (\mathcal{C}_\pi + \alpha\,\mathrm{Id}_{\mathcal{H}})^{-1/2}(\mathrm{Id}_{\mathcal{H}} - \mathcal{C}_\pi(\mathcal{C}_\pi + \alpha\,\mathrm{Id}_{\mathcal{H}})^{-1})I_\pi^* f_\star$$
$$= (\mathcal{C}_\pi + \alpha\,\mathrm{Id}_{\mathcal{H}})^{-1/2}(\mathrm{Id}_{\mathcal{H}} - \mathcal{C}_\pi(\mathcal{C}_\pi + \alpha\,\mathrm{Id}_{\mathcal{H}})^{-1})I_\pi^* \mathcal{T}_\pi^\nu f$
$$= I_\pi^*(\mathcal{T}_\pi + \alpha\,\mathrm{Id}_{L^2})^{-1/2}(\mathrm{Id}_{L^2} - \mathcal{T}_\pi(\mathcal{T}_\pi + \alpha\,\mathrm{Id}_{L^2})^{-1})\mathcal{T}_\pi^\nu f$$
$$= \tilde{U}\mathcal{T}_\pi^{1/2}(\mathcal{T}_\pi + \alpha\,\mathrm{Id}_{L^2})^{-1/2}(\mathrm{Id}_{L^2} - \mathcal{T}_\pi(\mathcal{T}_\pi + \alpha\,\mathrm{Id}_{L^2})^{-1})\mathcal{T}_\pi^\nu f,$$

for some $f \in L^2(\pi)$ satisfying $\|f\|_{L^2(\pi)} \leq R$ and the partial isometry $\tilde{U}: L^2(\pi) \to \mathcal{H}$ given by (4). This gives

$$\|I_{13}\|_{\mathcal{H}} \leq R \sup_{0 < t \leq \kappa^2} |\alpha t^{\nu+1/2}(t+\alpha)^{-3/2}|$$
$$\leq R \sup_{0 < t \leq \kappa^2} |\alpha t^\nu(t+\alpha)^{-1}| \sup_{0 < t \leq \kappa^2} |t^{1/2}(t+\alpha)^{-1/2}|$$
$$\leq R \sup_{0 < t \leq \kappa^2} |\alpha t^\nu(t+\alpha)^{-1}|$$
$$\leq R\max\{1, \kappa^{2(\nu-1)}\}\,\alpha^{\min\{\nu,1\}} = c_{13}\,\alpha^{\min\{\nu,1\}}, \tag{39}$$

where we use Lemma E.1 for the last inequality and set $c_{13} := R\max\{1, \kappa^{2(\nu-1)}\}$.

**Collecting all terms.** Collecting the bounds (37), (38), (39), together with (35) and taking a union bound gives with confidence $1 - \delta$

$$\|I_1\|_{L^2(\pi)} \leq 4(\|I_{11}\|_{\mathcal{H}} + \|I_{12}\|_{\mathcal{H}} + \|I_{13}\|_{\mathcal{H}})$$
$$\leq c_{\kappa,\nu,q,R}\left(\alpha^{\min\{\nu,1\}} + \frac{\log(4/\delta)}{\sqrt{\alpha}n} + \sqrt{\frac{\mathcal{N}(\alpha)\log(4/\delta)}{n}} + \sqrt{\mathcal{N}(\alpha)}\cdot\eta(\delta,n,\alpha)\right), \tag{40}$$

where we set

$$\eta(\delta, n, \alpha) := \max\left\{ \left(\frac{1}{\delta n^{q-1}}\right)^{1/q} \left(\frac{1}{\alpha \mathcal{N}(\alpha)}\right)^{\frac{q-2}{2q}}, \sqrt{\frac{\log(4c_1/\delta)}{n}} \right\},$$

and with $c_{\kappa,\nu,q,R} := 4\max\{c_{11}, c_{12}, c_{13}\}$.

### B.3.3  Bounding the second term $I_2$

We write again $\mathcal{C}_\alpha := \mathcal{C}_\pi + \alpha\,\mathrm{Id}_{\mathcal{H}}$, $\widehat{\mathcal{C}}_\alpha := \widehat{\mathcal{C}}_\pi + \alpha\,\mathrm{Id}_{\mathcal{H}}$ and find with Lemma E.2 and Corollary D.9 with confidence $1 - \delta$

$$
\begin{aligned}
\|I_2\|_{L^2(\pi)} &= \|\alpha I_\pi (\widehat{\mathcal{C}}_\pi + \alpha\,\mathrm{Id}_{\mathcal{H}})^{-1} f_\alpha\|_{L^2(\pi)} \\
&= \alpha\|I_\pi \mathcal{C}_\alpha^{-1/2} \mathcal{C}_\alpha^{1/2} \widehat{\mathcal{C}}_\alpha^{-1/2} \widehat{\mathcal{C}}_\alpha^{-1/2} \mathcal{C}_\alpha^{1/2} \mathcal{C}_\alpha^{-1/2} f_\alpha\|_{L^2(\pi)} \\
&= \alpha\|\mathcal{C}_\pi^{1/2} \mathcal{C}_\alpha^{-1/2} \mathcal{C}_\alpha^{1/2} \widehat{\mathcal{C}}_\alpha^{-1/2} \widehat{\mathcal{C}}_\alpha^{-1/2} \mathcal{C}_\alpha^{1/2} \mathcal{C}_\alpha^{-1/2} f_\alpha\|_{\mathcal{H}} \\
&\leq 4\alpha\, \|\mathcal{C}_\alpha^{-1/2} f_\alpha\|_{\mathcal{H}},
\end{aligned}
$$

where we again use (4) and $\|\mathcal{C}_\pi^{1/2} \mathcal{C}_\alpha^{-1/2}\|_{\mathcal{L}(\mathcal{H})} \leq 1$. Hence, by (SRC) and the definition of $f_\alpha$, we have with confidence $1 - \delta$

$$
\begin{aligned}
\|I_2\|_{L^2(\pi)} &\leq 4\alpha\, \|\mathcal{C}_\alpha^{-1/2} f_\alpha\|_{\mathcal{H}} \\
&\leq 4R \sup_{0 < t \leq \kappa^2} |\alpha(t + \alpha)^{-3/2} t^{\nu+1/2}| \\
&\leq 4R \max\{1, \kappa^{2(\nu-1)}\} \alpha^{\min\{\nu,1\}} = 4c_{13}\, \alpha^{\min\{\nu,1\}}, \tag{41}
\end{aligned}
$$

where we repeat the computation from (39).

### B.3.4  Final variance bound

Collecting now (33), (40) and (41) and taking a union bound gives the final bound for the variance. With confidence $1 - \delta$, we obtain

$$\|I_\pi(\widehat{f}_\alpha - f_\alpha)\|_{L^2(\pi)} \leq \tilde{c}_{\kappa,\nu,q,R} \left( \alpha^{\min\{\nu,1\}} + \frac{\log(8/\delta)}{\sqrt{\alpha}n} + \sqrt{\frac{\mathcal{N}(\alpha)\log(8/\delta)}{n}} + \sqrt{\mathcal{N}(\alpha)} \cdot \eta(\delta, n, \alpha) \right),$$

where we set

$$\eta(\delta, n, \alpha) := \max\left\{ \left(\frac{1}{\delta n^{q-1}}\right)^{1/q} \left(\frac{1}{\alpha \mathcal{N}(\alpha)}\right)^{\frac{q-2}{2q}}, \sqrt{\frac{\log(8c_1/\delta)}{n}} \right\},$$

and with $\tilde{c}_{\kappa,\nu,q,R} := 8\max\{c_{11}, c_{12}, c_{13}\}$.

### B.3.5  Bounding the bias $I_\pi f_\alpha - f_\star$

The bias can be bounded in the same way as in Section B.1.2:

$$\|I_\pi f_\alpha - f_\star\|_{L^2(\pi)} \leq R\alpha^{\min\{\nu,1\}}.$$

Combining the variance bound and the bias bound yields the result.

### B.4  Proof of Corollary 4.4

We first need a standard lemma that allows us to bound the effective dimension based on the eigenvalue decay of $\mathcal{T}_\pi$ due to [11, Lemma 11].

**Lemma B.1** (Eigenvalue decay)**.** *Assume* (EVD)*. Then there exists constant $\tilde{D} > 0$ such that*

$$\mathcal{N}(\alpha) \leq \tilde{D}\alpha^{-p} \quad \alpha > 0.$$

For simplicity, we will use the symbols $\lesssim$ for inequalities which hold up to a nonnegative multiplicative constant that does not depend on $n$ and $\delta$.

We first note that for every fixed $\delta \in (0,1)$, a short calculation shows that there exists an $\widetilde{n}_0 \in \mathbb{N}$ such that with the choice

$$\alpha(n,\delta) := \left( \frac{\log(8c_1/\delta)}{n} \right)^{\frac{1}{2\min\{\nu,1\}+p}}$$

the condition (14) is satisfied (since we have $n\alpha(n,\delta) \to \infty$ and $\sqrt{n}\alpha(n,\delta)^{(1+p)/2} \to \infty$ for $n \to \infty$). We may hence apply Proposition 4.3 and with confidence $1-\delta$, we have

$$\|I_\pi \widehat{f}_\alpha - f_\star\|_{L^2(\pi)} \le c \left( \alpha^{\min\{\nu,1\}} + \frac{\log(8/\delta)}{\sqrt{\alpha}n} + \sqrt{\frac{\mathcal{N}(\alpha)\log(8/\delta)}{n}} + \sqrt{\mathcal{N}(\alpha)} \cdot \eta(\delta,n,\alpha) \right),$$

(42)

with

$$\eta(\delta,n,\alpha) = \max\left\{ \left( \frac{1}{\delta n^{q-1}} \right)^{1/q} \cdot \left( \frac{1}{\alpha\mathcal{N}(\alpha)} \right)^{\frac{q-2}{2q}}, \sqrt{\frac{\log(8c_1/\delta)}{n}} \right\}$$

for all $n \ge \tilde{n}_0$ with a constant $c > 0$ not depending on $\delta$ and $n$.

We now show there exists $\overline{n}_0 \in \mathbb{N}$ such that

$$\sqrt{\mathcal{N}(\alpha(n,\delta))} \cdot \eta(\delta,n,\alpha(n,\delta)) \lesssim \sqrt{\frac{\log(8c_1/\delta)}{n\alpha(n,\delta)^p}}$$

(43)

for all $n \ge \overline{n}_0$, i.e., the subgaussian term asymptotically dominates the regularized Fuk–Nagaev term under the choice $\alpha(n,\delta)$. In fact, note that under $\mathcal{N}(\alpha) \le \tilde{D}\alpha^p$ ensured by Lemma B.1, we have

$$\sqrt{\mathcal{N}(\alpha)}\left( \frac{1}{\alpha\mathcal{N}(\alpha)} \right)^{\frac{q-2}{2q}} = \left( \frac{1}{\alpha} \right)^{\frac{q-2}{2q}} \mathcal{N}(\alpha)^{p/q} \le \left( \frac{1}{\alpha} \right)^{\frac{q-2}{2q}} \left( \frac{\tilde{D}}{\alpha} \right)^{p/q},$$

and we hence have

$$\sqrt{\mathcal{N}(\alpha)} \cdot \eta(\delta,n,\alpha) \lesssim \max\left\{ \left( \frac{1}{\delta n^{q-1}} \right)^{1/q} \cdot \left( \frac{1}{\alpha} \right)^{\frac{q-2}{2q}} \left( \frac{\tilde{D}}{\alpha} \right)^{p/q}, \sqrt{\frac{\log(8c_1/\delta)}{n\alpha^p}} \right\}. \quad (44)$$

We insert the definition of $\alpha = \alpha(n,\delta)$ into (44) and isolate the exponents corresponding to $1/n$ in both expressions inside of the above maximum. We obtain the exponent

$$\frac{q-1}{q} - \frac{1}{2\min\{\nu,1\}+p} \cdot \left( \frac{q-2}{2q} + \frac{p}{q} \right) = \frac{2(q-1)(2\min\{\nu,1\}+p) - (q-2+2p)}{2q(2\min\{\nu,1\}+p)} \quad (45)$$

for the first expression and

$$\frac{1}{2} - \frac{p/2}{2\min\{\nu,1\}+p} = \frac{\min\{\nu,1\}}{2\min\{\nu,1\}+p} = \frac{2q\min\{\nu,1\}}{2q(2\min\{\nu,1\}+p)} \quad (46)$$

for the second expression. We show that the difference between the numerator of (45) and the numerator of (46) is nonnegative. We have

$$2(q-1)(2\min\{\nu,1\}+p) - q + 2 - 2p - 2q\min\{\nu,1\} = (4q-4)\underbrace{\min\{\nu,1\}}_{\ge 1/2} + \underbrace{2p(q-2)}_{\ge 0} + 2 - q$$

$$\ge q - 2 + 2 - q = 0,$$

where we use $\nu \ge 2$, $q \ge 3$ and $p \in (0,1)$, hence proving (43).

Consequently, for $n \geq \max\{\widetilde{n}_0, \overline{n}_0\}$ and with Lemma B.1 and (43), the bound provided by (42) now reduces to

$$
\begin{aligned}
\|I_\pi \widehat{f}_{\alpha(n,\delta)} - f_\star\|_{L^2(\pi)} &\lesssim \alpha(n,\delta)^{\min\{\nu,1\}} + \frac{\log(8c_1/\delta)}{\sqrt{\alpha(n,\delta)n}} + \sqrt{\frac{\log(8c_1/\delta)}{n\alpha(n,\delta)^p}} \\
&\lesssim \left(\frac{\log(8c_1/\delta)}{n}\right)^{\frac{\min\{\nu,1\}}{2\min\{\nu,1\}+p}} + \left(\frac{\log(8c_1/\delta)}{n}\right)^{\frac{2\min\{\nu,1\}+p-1/2}{2\min\{\nu,1\}+p}} \\
&\quad + \left(\frac{\log(8c_1/\delta)}{n}\right)^{\frac{\min\{\nu,1\}}{2\min\{\nu,1\}+p}} \\
&\lesssim \left(\frac{\log(8c_1/\delta)}{n}\right)^{\frac{\min\{\nu,1\}}{2\min\{\nu,1\}+p}}
\end{aligned}
$$

with confidence $1 - \delta$, where we use $\sqrt{\log(8/\delta)} \leq \log(8/\delta) \leq \log(8c_1/\delta)$ since $c_1 \geq 1$, and bound the exponent of the middle term in the second step using $\nu \geq 1/2$.

## C  Excess risk bound for polynomial confidence regime

We now prove a simplified capacity-free excess risk bound based on Proposition 3.1 for the polynomial confidence regime $\delta \in D_2(n,q)$.

**Corollary C.1** (Polynomial confidence regime). *Let* (MOM) *and* (SRC) *be satisfied. There exists a constant $c > 0$ and a subset $\widetilde{D}_2(n,q) \subseteq D_2(n,q)$, defined in the proof, such that for all $\delta \in \widetilde{D}_2(n,q)$, we have*

$$
\|I_\pi \widehat{f}_{\alpha_2(n,\delta)} - f_\star\|_{L^2(\pi)} \leq c \cdot \alpha_2(n,\delta)^{\min\{\nu,1\}} \tag{47}
$$

*with confidence $1 - \delta$, where*

$$
\alpha_2(n,\delta) := \max\left\{ \left(\frac{1}{\delta n^{q-1}}\right)^{\frac{2}{q(2\min\{\nu,1\}+1)}}, \frac{\log(6/\delta)}{\sqrt{n}}, \kappa^2 \right\}.
$$

*Proof.* Let $\delta \in D_2(n,q)$. We have from Proposition 3.1 with confidence $1 - \delta$

$$
\|I_\pi \widehat{f}_\alpha - f_\star\|_{L^2(\pi)} \leq R\alpha^{\min\{\nu,1\}} + \frac{C_\diamond}{\sqrt{\alpha}}\left(\frac{\log(6/\delta)}{n} + \sqrt{\frac{\alpha^{2\min\{\nu,1\}}\log(6/\delta)}{n}} + \left(\frac{Q}{\delta n^{q-1}}\right)^{1/q}\right).
$$

Note that $\delta \in D_2(n,q)$ and $c_1 \geq 1$ ensure that

$$
\sqrt{\frac{\log(6/\delta)}{n}} \leq \sqrt{\frac{\log(6c_1/\delta)}{n}} \leq \frac{1}{\sigma}\left(\frac{Q}{\delta n^{q-1}}\right)^{1/q}.
$$

Hence, since $\alpha_2(n,\delta) \leq \kappa^2$, we find

$$
\sqrt{\frac{\alpha^{2\min\{\nu,1\}}\log(6/\delta)}{n}} \leq \frac{\kappa^{2\min\{\nu,1\}}}{\sigma}\left(\frac{Q}{\delta n^{q-1}}\right)^{1/q}.
$$

This leads to the bound

$$
\|I_\pi \widehat{f}_\alpha - f_\star\|_{L^2(\pi)} \leq R\alpha^{\min\{\nu,1\}} + \frac{C_\diamond}{\sqrt{\alpha}}\left(\frac{\log(6/\delta)}{n} + c_7\left(\frac{1}{\delta n^{q-1}}\right)^{1/q}\right),
$$

holding with confidence $1 - \delta$ and where we set $c_7 := Q^{1/q}\left(1 + \frac{\kappa^{2\min\{\nu,1\}}}{\sigma}\right)$.

Next, we observe that condition (9) implies

$$
\frac{\log(6/\delta)}{n} \leq \frac{1}{C_\kappa}\frac{\alpha}{\sqrt{n}}.
$$

Further assuming

$$\alpha \geq \left(\frac{1}{n}\right)^{\frac{1}{2\min\{\nu,1\}-1}},$$ (48)

then

$$\frac{C_\diamond}{C_\kappa \sqrt{\alpha}} \cdot \frac{\alpha}{\sqrt{n}} \leq \frac{C_\diamond}{C_\kappa} \alpha^{\min\{\nu,1\}} .$$

The upper bound for the excess risk reduces then to

$$\|I_\pi \widehat{f}_\alpha - f_\star\|_{L^2(\pi)} \leq c_8\, \alpha^{\min\{\nu,1\}} + \frac{c_9}{\sqrt{\alpha}} \left(\frac{1}{\delta n^{q-1}}\right)^{1/q}.$$

where we introduce $c_8 = R + \frac{C_\diamond}{C_\kappa}$ and $c_9 = c_7 C_\diamond$.

To ensure that the remaining variance part is of the same order as the approximation error part, we need to choose

$$\alpha \geq c_{10} \left(\frac{1}{\delta n^{q-1}}\right)^{\frac{2}{q(2\min\{\nu,1\}+1)}},$$ (49)

with $c_{10} = \left(\frac{c_9}{c_8}\right)^{\frac{2}{2\min\{\nu,1\}+1}}$. This finally gives with confidence $1 - \delta$

$$\|I_\pi \widehat{f}_\alpha - f_\star\|_{L^2(\pi)} \leq 2c_8\, \alpha^{\min\{\nu,1\}},$$

provided conditions $\delta \in D_2(n,q)$ (9), (48), (49) and $\alpha \leq \kappa^2$ hold. To simplify the conditions for $\alpha$, we observe that for all $q \geq 3$, $\nu \leq 1$ and $\delta \in (0,1)$ we have

$$\left(\frac{1}{n}\right)^{\frac{1}{2\min\{\nu,1\}-1}} \leq \left(\frac{1}{\delta n^{q-1}}\right)^{\frac{2}{q(2\min\{\nu,1\}+1)}}.$$

As a result, condition (49) implies (48) with an appropriate constant. To sum up, the regularization parameter needs to satisfy

$$\alpha \geq c_{11} \cdot \max\left\{\left(\frac{1}{\delta n^{q-1}}\right)^{\frac{2}{q(2\min\{\nu,1\}+1)}}, \frac{\log(6/\delta)}{\sqrt{n}}\right\},$$

with $c_{11} = \max\{1, c_{10}, C_\kappa\}$.

To ensure that $\alpha$ remains bounded by $\kappa^2$ it is sufficient to choose $n$ sufficiently large:

$$n \geq n_0(\delta) := c_{11} \cdot \max\left\{\kappa^{-2} \log^2(6/\delta), \delta^{-\frac{1}{q-1}} \kappa^{-\gamma}\right\},$$

with $\gamma = \frac{2q(2\min\{\nu,1\}+1)}{q-1}$. Recall that $\delta \in D_2(n,q)$ requires

$$n \leq n_{max}(\delta) := \left(\frac{Q^2}{\sigma^{2q}}\right)^{\frac{1}{q-2}} \delta^{-\frac{2}{q-2}} \cdot \log(6c_1/\delta)^{-\frac{q}{q-2}},$$

so we need to restrict $D_2(n,q)$ such that both conditions for $n$ are met. Letting now $\delta \in \widetilde{D}_2(n,q)$ with

$$\widetilde{D}_2(n,q) := \{\delta \in D_2(n,q) \ : \ n_0(\delta) \leq n_{max}(\delta)\},$$

then (47) holds with confidence $1 - \delta$.

$\square$

# D  Concentration bounds

We collect the concentration bounds used in the main text of this work and prove Proposition 5.1.

### D.1 Hoeffding inequality in Hilbert spaces

The following bound is classical and follows as a special case of [66, Theorem 3.5], see also [67, Section A.5.1].

**Proposition D.1** (Hoeffding inequality). *Let $\xi, \xi_1, \ldots \xi_n$ be independent random variables taking values in a Hilbert space $\mathcal{X}$ such that $\mathbb{E}[\xi] = 0$ and $\|\xi\|_{\mathcal{X}} \leq L$ almost surely. Then for all $t > 0$, we have*

$$\mathbb{P}\left[\left\|\frac{1}{n}\sum_{i=1}^{n}\xi_i\right\|_X \geq t\right] \leq 2\exp\left(-\frac{nt^2}{2L^2}\right).$$

### D.2 Bennett inequality in Hilbert spaces

We now give a version of a Bennett-type inequality going back to [66, Theorem 3.4]. We use the confidence bound version as derived by [4, Lemma 2].

**Proposition D.2** (Bennett inequality). *Let $\xi, \xi_1, \ldots \xi_n$ be independent and identically distributed random variables taking values in a Hilbert space $\mathcal{X}$ such that $\|\xi\|_{\mathcal{X}} \leq L$ almost surely and $\sigma^2 := \mathbb{E}[\|\xi\|_{\mathcal{X}}^2]$. Then for any $\delta \in (0, 1)$, we have*

$$\left\|\frac{1}{n}\sum_{i=1}^{n}\xi_i - \mathbb{E}[\xi]\right\|_{\mathcal{X}} \leq \frac{2L\log(2/\delta)}{n} + \sqrt{\frac{2\sigma^2\log(2/\delta)}{n}}$$

*with probability at least $1 - \delta$.*

### D.3 Fuk–Nagaev inequality in Hilbert spaces

The bound in Proposition 5.1 given in the main text is a sharper version of a more general bound given by [27] for general normed spaces, which we state here for completeness.

The original proof of the result below relies on an inconsistent exponential moment bound, which we address by providing an alternative bound alongside a detailed proof. We were made aware of this fact by Christian Fiedler [68], who contributed to the alternative proof presented here.

**Proposition D.3** (Fuk–Nagaev inequality, [27], Theorem 3.5.1). *Let $\xi_1, \ldots \xi_n$ be independent and identically distributed random variables taking values in a normed space $\mathcal{X}$ with measurable norm such that*

$$\mathbb{E}[\xi_i] = 0, \quad \sum_{i=1}^{n}\mathbb{E}[\|\xi_i\|_{\mathcal{X}}^2] < B^2 \quad and \quad \sum_{i=1}^{n}\mathbb{E}[\|\xi_i\|_{\mathcal{X}}^q] < A \tag{50}$$

*for some constants $B^2 > 0$, $A > 0$ and $q \in \mathbb{N}$, $q \geq 3$. Then there exist two universal constants $c_1 > 0$ and $c_2 > 0$ depending only on $q$, such that with $S_n := \sum_{i=1}^{n}\xi_i$ we have for every $t > 0$ that*

$$\mathbb{P}\big[\|S_n\|_{\mathcal{X}} - \mathbb{E}[\|S_n\|_{\mathcal{X}}] \geq tB\big] \leq c_1\left(\frac{A}{B^q t^q} + \exp(-c_2 t^2)\right).$$

Compared to the result above, Proposition 5.1 removes the excess term $\mathbb{E}[\|S_n\|_{\mathcal{X}}]$ on the left hand side by making use of the geometry of the Hilbert space norm (see [31, 66]).

We now prove Proposition 5.1 and emphasize that it can be directly extended to random variables in $(2, D)$-*smooth Banach spaces* by incorporating arguments from [66] with adjusted constants. We provide the proof in three parts: In Appendix D.3.1, we give an exponential moment bound. Appendix D.3.2 contains the core proof of Proposition 5.1 and incorporates the exponential moment bound into a Chernoff bound that is obtained via a truncation argument. Appendix D.3.3 provides the final optimization of the Chernoff bound.

#### D.3.1 Exponential moment bound

We now provide a sharpened version of [27, Lemma 3.5.1] that allows to bound the exponential moments of sums of random variables in Hilbert spaces. The proof of [27, Lemma 3.5.1] is lacking details and is inconsistent, as it generally seems to require bounded scaling factors $h > 0$ in the exponent, but the result is stated for all $h > 0$. Our proof holds for all $h > 0$ and gives a slightly adjusted bound.

**Lemma D.4.** *Let $\xi_1, \ldots \xi_n$ be independent random variables taking values in a separable Hilbert space $\mathcal{X}$ such that the conditions (50) are satisfied and additionally $\|\xi_i\|_{\mathcal{X}} \leq L$ holds almost surely for some $L > 0$ and all $1 \leq i \leq n$. Then for all $h > 0$, we have*

$$\mathbb{E}[\cosh(h\|S_n\|_{\mathcal{X}})] \leq \exp\big(\tilde{c}_2 h^2 B^2 + \tilde{c}_q A h^q e^{hL}\big),$$

*where the positive multiplicative constants $\tilde{c}_2$ and $\tilde{c}_q$ only depend on $q$.*

The upper bound of Lemma D.4 differs from [27, Lemma 3.5.1] in the sense that it removes an additive constant in the exponent. Furthermore, our constant $\tilde{c}_2$ of the quadratic term depends on $q$, while $\tilde{c}_2 = 1/2$ for all $q$ in [27, Lemma 3.5.1], which seems to conflict with unbounded $h > 0$.

*Proof.* We have

$$\mathbb{E}[\cosh(h\|S_n\|_{\mathcal{X}})] \leq \prod_{i=1}^{n} \mathbb{E}\Big[e^{h\|\xi_i\|_{\mathcal{X}}} - h\|\xi_i\|_{\mathcal{X}}\Big]$$

$$\leq \exp\bigg(\sum_{i=1}^{n} \mathbb{E}[e^{h\|\xi_i\|_{\mathcal{X}}} - 1 - h\|\xi_i\|_{\mathcal{X}}]\bigg)$$

where the first inequality is due to [31, Theorem 3], the second inequality follows from the fact that we have $x \leq \exp(x-1)$ for all $x \in \mathbb{R}$. We hence need to show

$$\sum_{i=1}^{n} \mathbb{E}[e^{h\|\xi_i\|_{\mathcal{X}}} - 1 - h\|\xi_i\|_{\mathcal{X}}] \leq \tilde{c}_1 h^2 B^2 + \tilde{c}_q A h^q e^{hL}.$$

We start by expanding the term on the left hand side as

$$\sum_{i=1}^{n} e^{h\|\xi_i\|_{\mathcal{X}}} - 1 - h\|\xi_i\|_{\mathcal{X}} = \sum_{i=1}^{n} \sum_{j=2}^{\infty} \frac{h^j \|\xi_i\|_{\mathcal{X}}^j}{j!}.$$

The term for the index $j = 2$ is immediately bounded as

$$\mathbb{E}\bigg[\sum_{i=1}^{n} \frac{h^2 \|\xi_i\|_{\mathcal{X}}^2}{2!}\bigg] \leq \frac{1}{2} h^2 B^2.$$

Hence, the claim now reduces to showing

$$\mathbb{E}\bigg[\sum_{i=1}^{n} \sum_{j=3}^{\infty} \frac{h^j \|\xi_i\|_{\mathcal{X}}^j}{j!}\bigg] \lesssim h^2 B^2 + A h^q e^{hL}$$

with constants only depending on $q$. We split the inner sum over $j$ at the index $q$ and obtain

$$\mathbb{E}\bigg[\sum_{i=1}^{n} \sum_{j=3}^{\infty} \frac{h^j \|\xi_i\|_{\mathcal{X}}^j}{j!}\bigg] = \mathbb{E}\bigg[\sum_{i=1}^{n} \sum_{j=3}^{q-1} \frac{h^j \|\xi_i\|_{\mathcal{X}}^j}{j!}\bigg] + \mathbb{E}\bigg[\sum_{i=1}^{n} \sum_{j=q}^{\infty} \frac{h^j \|\xi_i\|_{\mathcal{X}}^j}{j!}\bigg].$$

We now proceed to bound both terms on the right hand side individually.

**Bounding the first summand.** We address the first term as

$$\mathbb{E}\bigg[\sum_{i=1}^{n} \sum_{j=3}^{q-1} \frac{h^j \|\xi_i\|_{\mathcal{X}}^j}{j!}\bigg] \leq \mathbb{E}\bigg[\sum_{i=1}^{n} \sum_{j=3}^{q-1} \frac{\frac{j-2}{q-2} \cdot h^q \|\xi_i\|_{\mathcal{X}}^q + \frac{q-j}{q-2} \cdot h^2 \|\xi_i\|_{\mathcal{X}}^2}{j!}\bigg]$$

$$= \sum_{j=3}^{q-1} \frac{1}{j!} \sum_{i=1}^{n} \frac{j-2}{q-2} \cdot h^q \mathbb{E}[\|\xi_i\|_{\mathcal{X}}^q] + \frac{q-j}{q-2} \cdot h^2 \mathbb{E}[\|\xi_i\|_{\mathcal{X}}^2]$$

$$= \sum_{j=3}^{q-1} \frac{1}{j!} \bigg(\frac{j-2}{q-2} \cdot h^q A + \frac{q-j}{q-2} \cdot h^2 B^2\bigg)$$

$$\lesssim h^q A + h^2 B^2 \leq h^q A e^{hL} + h^2 B^2,$$

where we apply Lemma E.3 with $x = h\|\xi_j\|_{\mathcal{X}}$, $a = 1$ and $\rho = j$ to each individual term in the sum and use that $h > 0$ and $L > 0$ ensure $e^{hL} \geq 1$. Note that the unspecified constants only depend on $q$.

**Bounding the second summand.** It remains to bound the second sum. Almost surely, we have

$$
\begin{aligned}
\mathbb{E}\left[\sum_{i=1}^{n}\sum_{j=q}^{\infty}\frac{h^{j}\|\xi_{i}\|_{\mathcal{X}}^{j}}{j!}\right] &= \mathbb{E}\left[\sum_{i=1}^{n}\frac{h^{q}\|\xi_{i}\|_{\mathcal{X}}^{q}}{q!} + \frac{h^{q+1}\|\xi_{i}\|_{\mathcal{X}}^{q+1}}{(q+1)!}\cdots\right] \\
&= \mathbb{E}\left[\sum_{i=1}^{n}\frac{h^{q}\|\xi_{i}\|_{\mathcal{X}}^{q}}{q!}\left(1 + \frac{h\|\xi_{i}\|_{\mathcal{X}}}{(q+1)} + \frac{h^{2}\|\xi_{i}\|_{\mathcal{X}}^{2}}{(q+1)(q+2)}\cdots\right)\right] \\
&\leq \mathbb{E}\left[\sum_{i=1}^{n}\frac{h^{q}\|\xi_{i}\|_{\mathcal{X}}^{q}}{q!}\left(1 + \frac{h^{1}L}{(q+1)} + \frac{h^{2}L^{2}}{(q+1)(q+2)}\cdots\right)\right] \\
&\leq \frac{h^{q}}{q!}e^{hL}\sum_{i=1}^{n}\mathbb{E}[\|\xi_{i}\|_{\mathcal{X}}^{q}] \leq \frac{h^{q}}{q!}e^{hL}A,
\end{aligned}
$$

proving the claim. $\qquad\square$

### D.3.2   Core proof of Proposition 5.1

We can now prove Proposition 5.1 by modifying the truncation argument and Chernoff bound given by [27, Section 3.5.2] in combination with Lemma D.4. Let us assume that (50) holds. For $L > 0$, we introduce the truncated random variables

$$
\tilde{\xi}_{i} := \xi_{i}\,\mathbb{1}_{[\|\xi_{i}\|_{\mathcal{X}}\leq L]}, \qquad \tilde{S}_{n} := \sum_{i=1}^{n}\tilde{\xi}_{i}.
$$

In contrast to $S_{n}$, the truncated sum $\tilde{S}_{n}$ is not necessarily centered. Here, our version of the proof differs from the proof of [27, Theorem 3.5.1]. The original proof approximates the centered norm $\|S_{n}\|_{\mathcal{X}} - \mathbb{E}[\|S_{n}\|_{\mathcal{X}}]$ with the truncated centered norm $\|\tilde{S}_{n}\|_{\mathcal{X}} - \mathbb{E}[\|\tilde{S}_{n}\|_{\mathcal{X}}]$, leading to the excess term $\mathbb{E}[\|S_{n}\|_{\mathcal{X}}]$ in the event for which the tail is bounded. In contrast, we perform the centering directly in the norm: we approximate $\|S_{n}\|_{\mathcal{X}}$ with $\|\tilde{S}_{n} - \mathbb{E}[\tilde{S}_{n}]\|_{\mathcal{X}}$.

We first note that the difference between $\|\tilde{S}_{n}\|_{\mathcal{X}}$ and its centered version $\|\tilde{S}_{n} - \mathbb{E}[\tilde{S}_{n}]\|_{\mathcal{X}}$ can be bounded conveniently:

$$
\begin{aligned}
\left|\|\tilde{S}_{n}\|_{\mathcal{X}} - \|\tilde{S}_{n} - \mathbb{E}[\tilde{S}_{n}]\|_{\mathcal{X}}\right| &\leq \|\mathbb{E}[\tilde{S}_{n}]\|_{\mathcal{H}} = \|\mathbb{E}[\tilde{S}_{n}] - \mathbb{E}[S_{n}]\|_{\mathcal{X}} \\
&\leq \sum_{i=1}^{n}\mathbb{E}\left[\|\tilde{\xi}_{i} - \xi_{i}\|_{\mathcal{X}}\right] \\
&= \sum_{i=1}^{n}\mathbb{E}\left[\mathbb{1}_{[\|\xi_{i}\|_{\mathcal{X}}>L]}\|\xi_{i}\|_{\mathcal{X}}\right] \\
&= \sum_{i=1}^{n}\mathbb{E}\left[\mathbb{1}_{[\|\xi_{i}\|_{\mathcal{X}}>L]}\frac{\|\xi_{i}\|_{\mathcal{X}}^{q-1}}{\|\xi_{i}\|_{\mathcal{X}}^{q-1}}\|\xi_{i}\|_{\mathcal{X}}\right] \\
&\leq \frac{A}{L^{q-1}}.
\end{aligned} \tag{51}
$$

We now verify the conditions (50) for the centered sum $\tilde{S}_{n} - \mathbb{E}[\tilde{S}_{n}]$. We obviously have

$$
\sum_{i=1}^{n}\mathbb{E}[\|\tilde{\xi}_{i} - \mathbb{E}[\tilde{\xi}_{i}]\|_{\mathcal{X}}^{2}] \leq \sum_{i=1}^{n}\mathbb{E}[\|\tilde{\xi}_{i}\|_{\mathcal{X}}^{2}] \leq \sum_{i=1}^{n}\mathbb{E}[\|\xi_{i}\|_{\mathcal{X}}^{2}] \leq B^{2}.
$$

Furthermore, from Minkowski's inequality followed by $(a + b)^q \leq 2^{q-1}(a^q + b^q)$ for $a, b \geq 0$ and Jensen's inequality, we analogously obtain

$$\sum_{i=1}^{n} \mathbb{E}[\|\tilde{\xi}_i - \mathbb{E}[\tilde{\xi}_i]\|_{\mathcal{X}}^q] \leq \sum_{i=1}^{n} \left( \mathbb{E}[\|\tilde{\xi}_i\|_{\mathcal{X}}^q]^{1/q} + \mathbb{E}[\|\tilde{\xi}_i\|_{\mathcal{X}}] \right)^q$$

$$\leq 2^{q-1} \sum_{i=1}^{n} \left( \mathbb{E}[\|\tilde{\xi}_i\|_{\mathcal{X}}^q] + \mathbb{E}[\|\tilde{\xi}_i\|_{\mathcal{X}}]^q \right)$$

$$\leq 2^q \sum_{i=1}^{n} \mathbb{E}[\|\tilde{\xi}_i\|_{\mathcal{X}}^q] \leq 2^q \sum_{i=1}^{n} \mathbb{E}[\|\xi_i\|_{\mathcal{X}}^q] \leq 2^q A =: \tilde{A}.$$

We now provide the final tail bound for the norm of $S_n$ by approximating it with the centered truncated sum $\tilde{S}_n - \mathbb{E}[\tilde{S}_n]$. For every $t > 0$ and $h > 0$ we have

$$\mathbb{P}[\|S_n\|_{\mathcal{X}} \geq tB] \leq \mathbb{P}[S_n \neq \tilde{S}_n] + \mathbb{P}[\|\tilde{S}_n\|_{\mathcal{X}} \geq tB]$$

$$\leq \frac{A}{L^q} + \mathbb{P}[\|\tilde{S}_n\|_{\mathcal{X}} \geq tB] \qquad \text{(Markov's inequality)}$$

$$\leq \frac{A}{L^q} + \mathbb{P}[\|\tilde{S}_n - \mathbb{E}[\tilde{S}_n]\|_{\mathcal{X}} \geq tB - A/L^{q-1}] \qquad \text{(by (51))}$$

$$\leq \frac{A}{L^q} + \exp\left( -htB + \frac{hA}{L^{q-1}} \right) \mathbb{E}[\exp(h\|\tilde{S}_n - \mathbb{E}[\tilde{S}_n]\|_{\mathcal{X}})] \qquad \text{(Chernoff bound)}$$

$$\leq \frac{A}{L^q} + \exp\left( -htB + \frac{hA}{L^{q-1}} \right) 2\mathbb{E}[\cosh(h\|\tilde{S}_n - \mathbb{E}[\tilde{S}_n]\|_{\mathcal{X}})] \qquad (\cosh(x) = (e^x + e^{-x})/2)$$

$$\leq \frac{A}{L^q} + 2\exp\left( -htB + \frac{hA}{L^{q-1}} + \tilde{c}_2 h^2 B^2 + \tilde{c}_q \tilde{A} h^q e^{hL} \right) \qquad \text{(Lemma D.4)}$$

$$\leq 2\left( \frac{\tilde{A}}{L^q} + \exp\left( -htB + \frac{h\tilde{A}}{L^{q-1}} + \tilde{c}_2 h^2 B^2 + \tilde{c}_q \tilde{A} h^q e^{hL} \right) \right).$$

We derive an upper bound of the last expression in the parentheses over all choices of $L > 0$ and $h > 0$ as shown in detail in Appendix D.3.3, giving

$$\mathbb{P}\big[ \|S_n\|_{\mathcal{X}} \geq tB \big] \leq c_1 \left( \frac{\tilde{A}}{B^q t^q} + \exp(-c_2 t^2) \right) \tag{52}$$

for some positive constants $c_1$ and $c_2$ only depending on $q$. Note that we absorb the factor $2^q$ from $\tilde{A} = 2^q A$ into the generic constant $c_1$ in the resulting bound.

Whenever the $\xi_i$ satisfy $\mathbb{E}[\|\xi_i\|_{\mathcal{X}}^2] = \sigma^2$ and $\mathbb{E}[\|\xi_i\|_{\mathcal{X}}^q] = Q$ for some constants $\sigma^2, Q > 0$, we may substitute $B = n\sigma^2$ and $A = nQ$ in (52). Rearranging proves the bound given in Proposition 5.1.

### D.3.3   Chernoff bound optimization

The final bound we want to obtain is of the form

$$\min_{h,L \geq 0} \frac{\tilde{A}}{L^q} + \exp\left( -htB + \frac{h\tilde{A}}{L^{q-1}} + \tilde{c}_2 h^2 B^2 + \tilde{c}_q \tilde{A} h^q e^{hL} \right) \leq c_1 \left( \frac{\tilde{A}}{B^q t^q} + \exp(-c_2 t^2) \right) \tag{53}$$

with positive constants $c_1$ and $c_2$ only depending on $q$. We now adapt the idea of the original optimization argument by [27, pp. 105] to the setting of our Lemma D.4. We first introduce the shorthand notation

$$\Delta(t) := \tilde{A}/(B^q t^q) \quad \text{and} \quad \Lambda(t) := L/(tB).$$

With this notation, we rewrite the term on the left hand side of (53) as

$$\frac{\Delta(t)}{\Lambda(t)^q} + \exp\left( -htB + htB\Delta(t)/\Lambda(t)^{q-1} + \tilde{c}_2 h^2 B^2 + \tilde{c}_q (htB)^q \Delta(t) e^{\Lambda(t)thB} \right)$$

$$= \frac{\Delta(t)}{\Lambda(t)^q} + \exp\left( htB\big(\Delta(t)/\Lambda(t)^{q-1} - 1\big) + \tilde{c}_2 h^2 B^2 + \tilde{c}_q (htB)^q \Delta(t) e^{\Lambda(t)thB} \right). \tag{54}$$

Note that for all $t > 0$, $\Lambda(t)$ can be chosen independently of $\Delta(t)$ by adjusting the truncation level $L$ in our optimization. Note also that $\Delta(t)$ is exactly the first summand in the right hand side of the final desired bound (53). We will check the validity of the bound of the type (53) based on a case distinction.

**Large $\Delta(t)$.** We assume $\Delta(t) \geq 1$. For every $t > 0$, we may choose $L$ such that $\Lambda(t) > 1$ and proceed to bound (54) as

$$\frac{\Delta(t)}{\Lambda(t)^q} + \exp\left(htB\big(\Delta(t)/\Lambda(t)^{q-1} - 1\big) + \tilde{c}_2 h^2 B^2 + \tilde{c}_q (htB)^q \Delta(t) e^{\Lambda(t)thB}\right)$$

$$\leq \Delta(t) + \exp\left(htB(\Delta(t) - 1) + \tilde{c}_2 h^2 B^2 + \tilde{c}_q (htB)^q \Delta(t) e^{\Lambda(t)thB}\right)$$

$$\leq \Delta(t) + 2 \leq 3\Delta(t) + \exp(-c_2 t^2)$$

for any $c_2 > 0$, if we choose $h$ small enough such that

$$\exp(htB \underbrace{(\Delta(t) - 1)}_{\geq 0} + \tilde{c}_2 h^2 B^2 + \tilde{c}_q (htB)^q \Delta(t) e^{\Lambda(t)thB}) \leq 2.$$

This establishes (53).

**Small $\Delta(t)$.** We therefore need to consider the case where $\Delta(t) < 1$. Without loss of generality, in this setting we again assume that we are choosing $L$ appropriately to ensure that $\Lambda(t) > 1$. This gives us

$$\frac{\Delta(t)}{\Lambda(t)^q} \leq \Delta(t),$$

and hence we only need to bound the exponential term in (54), as the polynomial term satisfies the final bound (53). We distinguish two cases, as the exponential term in (54) behaves differently for different ratios of $t^2/\log(1/\Delta(t))$.

**Small $\Delta(t)$, case 1:** We assume $\Delta(t) < 1$, $\Lambda(t) > 1$ and $t^2 \leq 6\tilde{c}_2 \log(1/\Delta(t))$. We will show that in this case, the exponential term in (54) can be bounded by $c_1 \exp(-c_2 t^2)$.

We choose the optimization parameter $h := zt/B$ with some $z > 0$, which we will determine later. The exponent in the right hand side of (54) can now be written as

$$zt^2\big(\Delta(t)/\Lambda(t)^{q-1} - 1\big) + z^2 \tilde{c}_2 t^2 + \tilde{c}_q z^q t^{2q} \Delta(t) e^{\Lambda(t)zt^2}$$

$$= t^2\big(\Delta(t)/\Lambda(t)^{q-1} - z + \tilde{c}_2 z^2\big) + \tilde{c}_q z^q t^{2q} \Delta(t) e^{\Lambda(t)zt^2}$$

$$\leq t^2\big(z\Delta(t) - z + \tilde{c}_2 z^2\big) + \tilde{c}_q z^q (6\tilde{c}_2 \log(1/\Delta(t)))^q \Delta(t)^{1 - 6\tilde{c}_2 \Lambda(t)z}$$

$$\leq -\underline{c}_1 t^2 + 6^q \tilde{c}_q \tilde{c}_2^q z^q e^{-q} \left(\frac{q}{\underline{c}_2}\right)^q, \tag{55}$$

where we invoke Lemma E.4 in the last step and choose $z$ small enough such that we have

$$-z\Delta(t) + z - \tilde{c}_2 z^2 > c_1 > 0 \quad \text{and} \quad 1 - 6\tilde{c}_2 \Lambda(t)z > \underline{c}_2 > 0,$$

in which case the exponent estimate (55) is bounded by $-c_1 t^2 + \tilde{c}$, yielding the claim.

**Small $\Delta(t)$, case 2:** We assume $\Delta(t) < 1$, $\Lambda(t) > 1$ and $t^2 > 6\tilde{c}_2 \log(1/\Delta(t))$. We will show that in this case, the exponential term in (54) can be bounded by $c_1 \Delta(t)$.

In this regime, we choose the optimization parameter $h := z \log(1/\Delta(t))/(tB) > 0$ with some $z > 0$ which we will determine later. The exponent in (54) with this choice becomes

$$z \underbrace{\log(1/\Delta(t))}_{>0}(-1 + \Delta(t)/\Lambda(t)^{q-1}) + \tilde{c}_2 z^2 t^{-2} \underbrace{\log(1/\Delta(t))^2}_{<t^2 \log(1/\Delta(t))/(6\tilde{c}_2)}$$

$$+ \tilde{c}_q z^q \log(1/\Delta(t))^q \Delta(t) e^{\Lambda(t) z \log(1/\Delta(t))}$$

$$\le z \log(1/\Delta(t))(-1 + \Delta(t)) + \frac{z^2}{6} \log(1/\Delta(t)) + \tilde{c}_q z^q \log(1/\Delta(t))^q \Delta(t)^{1-6\tilde{c}_2 \Lambda(t) z}$$

$$= \underbrace{\log(\Delta(t))}_{<0}(z - \Delta(t) z - \frac{z^2}{6}) + \tilde{c}_q z^q \log(1/\Delta(t))^q \Delta(t)^{1-6\tilde{c}_2 \Lambda(t) z}$$

$$\le \underbrace{\log(\Delta(t))}_{<0} + \tilde{c}_q z^q e^{-q} \left(\frac{q}{\underline{c}}\right)^q, \qquad (56)$$

where in the last step we apply Lemma E.4 and choose $z$ small enough such that

$$z - \Delta(t) z - \frac{z^2}{6} \ge 1 \quad \text{and} \quad 1 - 6\tilde{c}_2 \Lambda(t) z > \underline{c} > 0.$$

We finally obtain an upper bound of the exponent (56) as $\log(\Delta(t)) + \tilde{c}$, yielding the claim.

**Final bound:**  The final bound (53) results from considering all of the the above cases and choosing the individual constants $c_1$ and $c_2$ large enough such that all cases are satisfied simultaneously (for example by choosing the maximum of the individual constants obtained above).

### D.4 Concentration of empirical covariance operators

We have the following classical bound on the estimation error of the empirical covariance operators when Proposition D.1 is applied to the centered random operators $\xi_i := \phi(X_i) \otimes \phi(X_i) - \mathcal{C}_\pi$ which are bounded by $2\kappa^2$ in Hilbert–Schmidt norm under Assumption 2.1. Note that the operator norm can be bounded by the Hilbert–Schmidt norm.

**Corollary D.5** (Sample error of empirical covariance operator)**.** *Under Assumption 2.1, we have*

$$\left\|\widehat{\mathcal{C}}_\pi - \mathcal{C}_\pi\right\|_{S_2(\mathcal{H})} \le 2\kappa^2 \sqrt{\frac{2\log(2/\delta)}{n}}$$

*with probability at least $1 - \delta$.*

We now recall a result by [69, Proposition 1].

**Proposition D.6** (Empirical inverse, [69], Proposition 1)**.** *Let Assumption 2.1 hold. For all $\delta \in (0,1), n \in \mathbb{N}, \alpha > 0$, denote*

$$\mathcal{B}(n, \delta, \alpha) := 1 + \log^2(2/\delta)\left(\frac{2\kappa^2}{n\alpha} + \sqrt{\frac{4\kappa^2 \mathcal{N}(\alpha)}{n\alpha}}\right)^2. \qquad (57)$$

*With probability at least $1 - \delta$*

$$\left\|(\widehat{\mathcal{C}}_\pi + \alpha \operatorname{Id}_\mathcal{H})^{-1}(\mathcal{C}_\pi + \alpha \operatorname{Id}_\mathcal{H})\right\|_{\mathcal{L}(\mathcal{H})} \le 2\mathcal{B}(n, \delta, \alpha). \qquad (58)$$

*In particular, if*

$$\frac{1}{n\alpha} \le \mathcal{N}(\alpha), \qquad 16 \max\{1, \kappa^4\} \log^2(2/\delta) \frac{\mathcal{N}(\alpha)}{\alpha} \le n, \qquad (59)$$

*then $\left\|(\widehat{\mathcal{C}}_\pi + \alpha \operatorname{Id}_\mathcal{H})^{-1}(\mathcal{C}_\pi + \alpha \operatorname{Id}_\mathcal{H})\right\|_{\mathcal{L}(\mathcal{H})} \le 4$.*

*Proof.* We prove the particular case. Note that if $(n\alpha)^{-1} \le \mathcal{N}(\alpha)$,

$$\mathcal{B}(n, \delta, \alpha) \le 1 + \log^2(2/\delta)(\kappa + 1)^2 \frac{4\kappa^2 \mathcal{N}(\alpha)}{n\alpha} \le 1 + 16 \max\{1, \kappa^4\} \log^2(2/\delta) \frac{\mathcal{N}(\alpha)}{n\alpha}.$$

Therefore, if $16 \max\{1, \kappa^4\} \log^2(2/\delta)\mathcal{N}(\alpha) \le n\alpha$, then $\mathcal{B}(n, \delta, \alpha) \le 2$. $\qquad \square$

We next give a Corollary, providing a simplified bound under a worst case assumption for the effective dimension that can be applied without information about the eigenvalue decay.

**Corollary D.7** (Empirical inverse, worst case scenario). *Let Assumption 2.1 hold. Assume that $\mathcal{N}(\alpha) \leq c\alpha^{-1}$ for some $c > 0$. Then, with confidence $1 - \delta$, we have*

$$\left\| (\widehat{\mathcal{C}}_\pi + \alpha \operatorname{Id}_{\mathcal{H}})^{-1} (\mathcal{C}_\pi + \alpha \operatorname{Id}_{\mathcal{H}}) \right\|_{\mathcal{L}(\mathcal{H})} \leq 4,$$

*provided that*

$$C_\kappa \frac{\log(2/\delta)}{\alpha} \leq \sqrt{n}, \qquad C_\kappa := 2(1 + \sqrt{c}) \cdot \max\{1, \kappa^2\} .$$

*Proof.* Using $\mathcal{N}(\alpha) \leq c\alpha^{-1}$ and plugging it into (57), we obtain

$$\mathcal{B}(n, \delta, \alpha) \leq 1 + \log^2(2/\delta) \left( \frac{2\kappa^2}{n\alpha} + \frac{2\sqrt{c}\kappa}{\sqrt{n\alpha}} \right)^2 .$$

Note that

$$\frac{2\kappa^2}{n\alpha} + \frac{2\sqrt{c}\kappa}{\sqrt{n\alpha}} \leq \frac{2(1 + \sqrt{c}) \cdot \max\{1, \kappa^2\}}{\sqrt{n\alpha}} = \frac{C_\kappa}{\sqrt{n\alpha}}.$$

Therefore,

$$\mathcal{B}(n, \delta, \alpha) \leq 1 + \log^2(2/\delta) \cdot \frac{C_\kappa^2}{n\alpha^2}.$$

To ensure $\mathcal{B}(n, \delta, \alpha) \leq 2$, it suffices that

$$\log^2(2/\delta) \cdot \frac{C_\kappa^2}{n\alpha^2} \leq 1 \quad \Longleftrightarrow \quad C_\kappa \cdot \frac{\log(2/\delta)}{\alpha} \leq \sqrt{n}.$$

Under this condition, we obtain $\mathcal{B}(n, \delta, \alpha) \leq 2$, and hence

$$\left\| (\widehat{\mathcal{C}}_\pi + \alpha I)^{-1} (\mathcal{C}_\pi + \alpha I) \right\| \leq 4,$$

as claimed. $\qquad\square$

**Remark D.8** (Worst case scenario). Note that, by Assumption 2.1, we always have

$$\mathcal{N}(\alpha) \leq \operatorname{tr}(\mathcal{C}_\pi)\alpha^{-1} = \mathbb{E}[\|\phi(X)\|_{\mathcal{H}}^2]\alpha^{-1} \leq \kappa^2\alpha^{-1}.$$

We therefore see that the constant $c$ in the above assumption always exists and satisfies $c \leq \kappa^2$.

Under the eigenvalue decay assumption (EVD), the conditions of Proposition D.6 can be simplified.

**Corollary D.9** (Empirical inverse under (EVD)). *Let Assumption 2.1 and (EVD) hold. Then, with confidence $1 - \delta$, we have*

$$\left\| (\widehat{\mathcal{C}}_\pi + \alpha \operatorname{Id}_{\mathcal{H}})^{-1} (\mathcal{C}_\pi + \alpha \operatorname{Id}_{\mathcal{H}}) \right\|_{\mathcal{L}(\mathcal{H})} \leq 4,$$

*provided that*

$$\log(2/\delta) \left( \frac{2\kappa^2}{n\alpha} + \frac{2\sqrt{\tilde{D}}\kappa}{\sqrt{n}\alpha^{(1+p)/2}} \right) \leq 1,$$

*where $\tilde{D}$ is defined in Lemma B.1.*

*Proof.* By Lemma B.1, under (EVD), there exists a constant $\tilde{D} > 0$ such that $\mathcal{N}(\alpha) \leq \tilde{D}\alpha^{-p}$. Inserting this bound into (57) leads to

$$\mathcal{B}(n, \delta, \alpha) \leq 1 + \log^2(2/\delta) \left( \frac{2\kappa^2}{n\alpha} + \frac{2\sqrt{\tilde{D}}\kappa}{\sqrt{n}\alpha^{(1+p)/2}} \right)^2 .$$

$\qquad\square$

We can use Proposition D.6 to bound the weighted norm of a function in $\mathcal{H}$ by an empirically weighted and regularized norm.

**Lemma D.10** (Concentration of weighted regularized norm)**.** *Under Assumption 2.1, for all $f \in \mathcal{H}$ and all $\alpha > 0$ and $s \in [0, 1]$, we have*

$$\|\mathcal{C}_\pi^s f\|_{\mathcal{H}} \leq 2^s \mathcal{B}(n, \delta, \alpha)^s \left\|(\widehat{\mathcal{C}}_\pi + \alpha \operatorname{Id}_{\mathcal{H}})^s f\right\|_{\mathcal{H}} \tag{60}$$

*with probability at least $1 - \delta$.*

*Proof.* We have

$$\|\mathcal{C}_\pi^s f\|_{\mathcal{H}} \leq \left\|\mathcal{C}_\pi^s (\mathcal{C}_\pi + \alpha \operatorname{Id}_{\mathcal{H}})^{-s}\right\|_{\mathcal{L}(\mathcal{H})} \left\|(\mathcal{C}_\pi + \alpha \operatorname{Id}_{\mathcal{H}})^s (\widehat{\mathcal{C}}_\pi + \alpha \operatorname{Id}_{\mathcal{H}})^{-s}\right\|_{\mathcal{L}(\mathcal{H})} \left\|(\widehat{\mathcal{C}}_\pi + \alpha \operatorname{Id}_{\mathcal{H}})^s f\right\|_{\mathcal{H}}. \tag{61}$$

We bound the terms on the right hand side individually. Applying Lemma E.2 to the first term on the right hand side of (61), we have

$$\left\|\mathcal{C}_\pi^s (\mathcal{C}_\pi + \alpha \operatorname{Id}_{\mathcal{H}})^{-s}\right\|_{\mathcal{L}(\mathcal{H})} \leq \left\|\mathcal{C}_\pi (\mathcal{C}_\pi + \alpha \operatorname{Id}_{\mathcal{H}})^{-1}\right\|_{\mathcal{L}(\mathcal{H})}^s \leq 1.$$

The second term can be bounded with Proposition D.6. $\qquad\square$

# E  Miscellaneous results

We recall a standard result that addresses the saturation property of Tikhonov regularization, see e.g. [59, Example 4.15].

**Lemma E.1.** *Let $\alpha > 0$ and $\nu > 0$. We have*

$$\sup_{t \in [0, \kappa^2]} \frac{\alpha t^\nu}{t + \alpha} \leq \max\{1, \kappa^{2(\nu-1)}\} \alpha^{\min\{\nu, 1\}}.$$

We will also frequently use the following classical bound for products of fractional operators.

**Lemma E.2** (Cordes inequality, [70])**.** *Let $s \in [0, 1]$ and $A, B \in \mathcal{L}(\mathcal{H})$ be positive-semidefinite selfadjoint. Then we have $\|A^s B^s\|_{\mathcal{L}(\mathcal{H})} \leq \|AB\|_{\mathcal{L}(\mathcal{H})}^s$.*

The next result is part of [40, Proposition 3.5]. It allows us to bound moments between 2 and $q$. For convenience, we provide a detailed, self-contained proof.

**Lemma E.3.** *Let $q > 2$. For all $\rho \in [2, q]$, $a \in \mathbb{R}_{>0}$, and $x \in \mathbb{R}_{>0}$ we have*

$$(q - 2)x^\rho \leq (\rho - 2)a^{\rho-q}x^q + (q - \rho)a^{\rho-2}x^2. \tag{62}$$

*Proof.* We first show that

$$(q - 2)\left(\frac{x}{a}\right)^{\rho-2} \leq (\rho - 2)\left(\frac{x}{a}\right)^{q-2} + (q - \rho). \tag{63}$$

For this, define for some $y \in \mathbb{R}_{>0}$ and $t \in \mathbb{R}_{>0}$ the function $f : [0, t] \to \mathbb{R}$ by $f(s) = \exp(\ln(y)s)$. Note that $f$ is convex (since $f'(s) = f(s)\ln(y)$, $f''(s) = f(s)\ln(y)^2 > 0$). For $s \in [0, t]$ we then get

$$
\begin{aligned}
y^s = f(s) &= f((1 - s/t) \cdot 0 + s/t \cdot t) \\
&\leq (1 - s/t)f(0) + s/t \cdot f(t) = (1 - s/t) + s/t \cdot y^t
\end{aligned}
$$

and multiplying both sides by $t$ leads to

$$t y^s \leq s y^t + (t - s).$$

Setting $t = q - 2$, $s = \rho - 2$, and $y = x/a$ then shows (63). Finally, multiplying both sides of this with $x^2/a^{2-\rho}$ establishes the result. $\qquad\square$

Finally, we require the following basic inequality.

**Lemma E.4.** *Consider the function* $f : (0,1) \to \mathbb{R}_+$ *given by* $f(\alpha) = \log(1/\alpha)^q \alpha^s$ *for some* $s > 0$ *and* $q > 0$. *Then we have*

$$\sup_{\alpha \in (0,1)} f(\alpha) \leq e^{-q} \left(\frac{q}{s}\right)^q.$$

*Proof.* We may substitute the transformation $x := \log(1/\alpha) \in (0, \infty)$ and consider $f(x) = x^q \exp(-sx)$ instead. Note that $f$ is nonnegative on $(0, \infty)$ and that $f(0) = \lim_{x \to \infty} f(x) = \lim_{x \to 0_+} f(x) = 0$. The derivative $f'(x) = \exp(-sx)(qx^{q-1} - sx^q)$ has the unique root $x_\star = q/s$ and we hence obtain $f(x_\star) = \exp(-q)(q/s)^q$ as the maximal value. $\square$

