# OpenReview forum: "Regularized least squares learning with heavy-tailed noise is minimax optimal"
_NeurIPS.cc/2025/Conference — NeurIPS 2025 spotlight_

### Official Review · Reviewer_2L1C · 2025-06-24

**Clarity:** 3
**Significance:** 3
**Originality:** 2
**Rating:** 5
**Confidence:** 4

**Summary:**

This paper studies the generalization performance of ridge regression under heavy-tailed noises,
showing optimal rates similar to that derived unde subexponential noises in the literature.

**Questions:**

1. Can you discuss the requirement of $q \geq 3$ in Assumption 2.2? Can it be improved to $q > 2$?

2. As mentioned in the paper, the typical assumption that the kernel is bounded leads to subgaussian concentration of the embedded covariates. Can the techiques in this paper be further applied to "heavy-tailed" kernels that is unbounded?

**Ethical Concerns:**

["NO or VERY MINOR ethics concerns only"]

**Final Justification:**

The theory in this paper is solid and additional improvements are made during revision.
I would like keep my opinion to accept this paper.

**Limitations:**

yes

**Quality:**

3

**Strengths And Weaknesses:**

## Strengths

* This paper is well written and easy to read.
* This paper extends the analysis of kernel ridge regression under general heavy-tailed noise, which is novel.  By showing the same optimal convergence rate as the sub-Gaussian noise case, the paper also shows the robustness of regularized least squares against heavy-tailed noise.
* In terms of technical contribution, the application of Fuk–Nagaev inequality in this paper can also be useful in other kernel-related problems.

## Weakness

* Numerical simulations could be provided to validate the theoretical results and further demonstrate the methods robustedness, particularly for noises with different tail behaviors.

* This paper does not provide practical ways of choosing the hyperparameter $\alpha$ under different confidence levels, limiting the paper's real-world implications.

---

> ### Author Rebuttal · Authors · 2025-07-28
>
> Thank you for the effort in reviewing our manuscript and the very encouraging feedback. We incorporated insights based on your questions and comments into our manuscript.
>
> Additionally, we believe you pointed out *two important directions for future work* (empirical parameter choice rules and learning rates for unbounded kernels), which we think are now within reach based on the theory and tools we provide in the paper. In fact, we spent some time thinking about these ideas during the writing of this manuscript and highlight some of our ideas alongside general answers to your questions below.
>
> **Empirical experiments** (as highlighted by the other reviewers)
>
> We definitely agree that basic numerical simulations confirming the behavior of our upper bounds will improve the paper.
> In fact, as outlined in detail in our reply to **Reviewer AM9J**, we have completed a series of experiments shortly after the submission and compared the empirical distributions of the excess risks of the heavy-tailed and the light-tailed setting. Our theoretical results are fully confirmed by these experiments and we present them in the final version of the manuscript using the additional space.
>
> **Hyperparameter choice $\alpha(n, \delta)$**
>
> We agree with the reviewer again. However, we see our work as a starting point for more practical results for kernel regression with heavy-tailed noise in the sense that we first answer the fundamental question *"What is the optimal rate and how does the optimal (theoretical) parameter choice look like"?*
>
> Based on our results, empirical parameter choice rules can now be derived analogously to the light-tailed setting based on the results and approach presented in the paper.
> A specific empirical parameter choice rule stemming from inverse problems is *Morozov's discrepancy principle* (e.g. [1]),
> which can be adapted to the noisy learning setting [2]. We plan to investigate this in future work, specifically also in the context of the polynomial confidence regime, which differs from the light-tailed setting. We added a corresponding remark on empirical parameter choice rules to the manuscript after the we introduce the optimal parameter choice for the capacity-free setting to make the reader aware of this fact. In particular, asymptotically as $n \to \infty$, rate optimal rules for the light-tailed setting are optimal for the heavy-tailed setting as well as demonstrated by our results. However, the empirical estimation of the involved quantities for the practical construction of these rules might be impacted by heavy-tailed noise, which is yet to be investigated.
>
> **Extension to $q> 2$**
>
> Currently, the restriction $q \geq 3$ stems from the assumption of the Fuk-Nagaev inequality, the rest of our proofs is compatible with the case $q> 2$. In particular, we believe that the original truncation argument used by [3] for the bound in normed spaces is suboptimal and the bound can be improved to $q > 2$ (as is the case for the original real-valued bound due to Fuk and Nagaev). We refer the reviewer also to our more detailed reply to **Reviewer AM9J**, who asked the same important question.
>
> **Unbounded kernels**
>
> This is another good point which we believe can be tackled with our approach. Currently, our results rely on the fact that due to boundedness of the kernel, the concentration of the empirical covariance operator can be bounded conveniently based on the Bennett inequality. However, only assuming $E [k(X,X)^{s}] < \infty$ for $s \geq 3$ (or $s > 2$ given the discussion above) and applying the Fuk-Nagaev bound a second time seems possible as well. This step leads to a final bound with *two polynomial terms*, one based on $s$ (operator perturbation) and one based on $q$ (model noise). Incorporating similar arguments as in our current proofs to the additional polynomial term might lead to optimal rates as well. We believe this could be a promising first step towards general learning rates for unbounded kernels.
>
>
> **References**
>
> [1] H. W. Engl, M. Hanke, A. Neubauer. Regularization of Inverse Problems (1996).
>
> [2] A. Celisse, M. Wahl. Analyzing the discrepancy principle for kernelized spectral filter learning algorithms. JMLR (2021).
>
> [3] V. Yurinsky. Sums and Gaussian vectors (1995).

---

> > ### Comment · Reviewer_2L1C · 2025-08-01
> >
> > Thank you for the response. I will keep my score.

---

### Official Review · Reviewer_rSYV · 2025-06-29

**Clarity:** 3
**Significance:** 3
**Originality:** 3
**Rating:** 5
**Confidence:** 3

**Summary:**

The paper analyzes the performance of ridge regression in reproducing kernel Hilbert spaces (RKHS) when the observational noise has heavy tails, meaning it lacks exponential tail decay. Unlike prior work assuming subexponential noise (e.g., satisfying a Bernstein condition), the authors only require a finite number of conditional moments. They prove that ridge regression remains statistically robust under these weaker assumptions by using a Hilbert-space version of the Fuk–Nagaev inequality. They show that the convergence rates match known optimal rates under standard eigenvalue decay assumptions, thereby establishing that regularized least squares is minimax optimal even under heavy-tailed noise. The work generalizes existing theory and demonstrates that tight learning guarantees are achievable without strong tail assumptions, making kernel methods more broadly applicable in practice.

**Questions:**

- Have you considered or do you plan to validate your theoretical results with empirical experiments — particularly comparing performance under subgaussian vs. heavy-tailed noise?

- The choice of the squared loss is not deeply discussed in the paper. Do you think your analysis—particularly the use of Fuk–Nagaev inequalities in Hilbert spaces—can be extended to learning problems with general convex loss functions? For instance, would similar results be attainable for regularized learning algorithms such as SVMs or other kernel-based methods involving Lipschitz or piecewise-linear losses?

**Ethical Concerns:**

["NO or VERY MINOR ethics concerns only"]

**Final Justification:**

I confirm, even more after the rebuttal, that this paper deserve in my opinion to be accepted.

**Limitations:**

yes

**Quality:**

3

**Strengths And Weaknesses:**

Strengths: the paper is clear and well structured, results are explained and motivated precisely. The assumed setting is meaningful and tries to solve an interesting problem. The paper brings technical novelty to widespread ridge regression and the level of technical sophistication of the tools and analysis is high.

Weaknesses: no important ones. Despite the paper being clearly theoretical, it would have been interesting to see some practical application or simulation under this heavy-tailed noise.

---

> ### Author Rebuttal · Authors · 2025-07-28
>
> We thank the reviewer for their for their positive and encouraging evaluation of our work, their time spent in reviewing our manuscripts well as their helpful feedback. We comment on your two questions below.
>
> **Empirical experiments**
>
> We evaluated a series of numerical simulations shortly after the submission as pointed out in full detail in the reply to **Reviewer AM9J**. In particular, we compare the empirical distributions of the excess risks for different parameter combinations (sample size, regularization parameter, distributional assumptions) in a light-tailed setting and a heavy-tailed setting. Our theoretical results are fully confirmed by these experiments: one can clearly distinguish the subgaussian confidence regime (corresponding quantiles of excess risk distributions of light-tailed and heavy-tailed models overlap) and the polynomial regime (quantiles of the excess risk distributions of the heavy-tailed model are larger). We present experiments in the final version of the manuscript using the additional space.
>
> **Other loss functions**
>
> We agree with the reviewer that the implications of the choice of squared loss and the extension to other losses should be highlighted. We also see that this may lead to important future work. We added a remark to the manuscript that outlines the potential application of our ideas to other loss functions (also see the reply to **Reviewer JkXL** for a more detailed comparion of results based on bounded noise) and outline a potential direction below.
>
> As an alternative to the integral operator approach, an *empirical process approach* has been proposed (see e.g. [1] for an overview) for kernel machines with more general losses. This approach can incorporate the eigenvalue decay to derive optimal rates [2], usually relying on boundedness or Lipschitz assumptions of the loss under the noise. In order to obtain optimal rates with unbounded (and in particular heavy-tailed) noise, these arguments might need to be combined with a (uniform) Fuk-Nagaev inequality over a ball of RKHS functions (potentially with a truncated loss, as demonstrated in the proof of the FN-inequality). As the empirical process combines the eigenvalue decay of the integral operator with an entropy argument, we believe that *repeating the trick for the polynomial excess risk component* as presented in our paper might generally be possible by replacing the effective dimension with a suitable entropy proxy. We hope that our work motivates the extension of the empirical process approach to unbounded and heavy-tailed losses in the future.
>
> [1] I. Steinwart and A. Christmann. Support Vector Machines (2008).
>
> [2] I. Steinwart, D.R. Hush, C. Scovel. Optimal rates for regularized least squares regression. COLT (2009).

---

> > ### Comment · Reviewer_rSYV · 2025-08-04
> >
> > I thank the authors for their detailed response. I intend to keep my rating.

---

### Official Review · Reviewer_JkXL · 2025-07-02

**Clarity:** 3
**Significance:** 4
**Originality:** 3
**Rating:** 5
**Confidence:** 4

**Summary:**

This paper investigates the performance of ridge regression in reproducing kernel Hilbert spaces under heavy-tailed noise conditions, where the noise has only finite higher moments. The authors derive excess risk bounds for ridge regression consisting of subgaussian and polynomial terms, demonstrating that the dominant subgaussian term allows for convergence rates previously only achieved under subexponential noise assumptions. The results show that the derived rates are minimax optimal under standard eigenvalue decay conditions, proving the robustness of regularized least squares against heavy-tailed noise. The analysis relies on a Fuk-Nagaev inequality for Hilbert-space valued random variables.

**Questions:**

1.	Why isn’t least squares commonly used in practice under heavy-tailed conditions? Providing some insightful remarks would be beneficial.

2.	How do the derived rates compare to those of robust regression methods (e.g., Huber loss) under the same noise conditions?

3.	The distinction between subgaussian and polynomial regimes is insightful. Are there practical guidelines for choosing the regularization parameter in each regime?

**Ethical Concerns:**

["NO or VERY MINOR ethics concerns only"]

**Final Justification:**

I thank the authors for addressing my previous concerns. This paper offers valuable insights and contributes significantly to the field of robust learning, and I suggest its acceptance.

**Limitations:**

yes

**Quality:**

4

**Strengths And Weaknesses:**

Strengths

1. The use of the Fuk-Nagaev inequality is innovative and well-justified.

2. The results cover both capacity-independent and capacity-dependent settings, offering a comprehensive view of the problem.

3. The work has significant implications for robust statistical learning, as it shows that optimal rates can be achieved even with heavy-tailed noise.

Weaknesses

1. While the theoretical contributions are strong, the paper lacks empirical validation.

2. The comparison to related work may lack sufficient detail (e.g. [37-38]) and may be unclear to non-experts.

3. A discussion of practical implications is absent. How do the results inform the choice of loss functions in the presence of heavy-tailed noise?

---

> ### Author Rebuttal · Authors · 2025-07-29
>
> Thank you very much for the review of our manuscript, the insightful comments and the encouraging endorsement of our work.
>
> We have addressed all of your points and edited the manuscript accordingly. We comment on each individual point below.
> Please note that we also refer to our replies to the other reviewers, as we did not want to redundantly list details and references to existing work regarding similar questions.
>
> **Empirical validation**
>
> We agree. Just after the submission of this paper, we completed numerical evaluations which compare heavy-tailed and light-tailed models in terms of their quantile functions of the excess risks.
> The results fully align with the excess risk distributions described by our upper bounds and confidence regimes. Please refer to our answer to **Reviewer AM9J** for the details and the setup of the empirical experiments.
> The experiments will be included in the final paper using the additional space.
>
> **Comparison to other results, rates and losses**
>
> We added an extended discussion of kernel machines with other losses in the introduction. We already briefly touched upon robust kernel regression with the Cauchy loss [1], for which slower rates than ours have been derived. However, it is very important to note that the Cauchy loss can be applied to the scenario $q<2$, in which case the squared loss is undefined and the general integral operator approach with squared loss does not apply.
>
> For general convex and Lipschitz losses (including Huber loss), we are aware of the family of results described in Chapters 6 & 7
> of [2], which derive rates under boundedness/loss clipping assumptions *not covering heavy-tailedness of the losses*. These rates are are generally not optimal (see the discussion of the rates [2, p. 268] and a related the derivation of optimal rates for squared loss with bounded noise [3]). Please note that the so-called *approximation function* used in this context can be directly related to our source condition [3]. Additionally, the rates in the capacity-independent setting are generally slower than ours either by a factor in the exponent or a logarithmic term in $n$. From our viewpoint, our work motivates the derivation of optimal rates for these losses *without clipping or bounded noise assumptions* by combining the above approaches with similar applications of a Fuk-Nagaev type inequality, as no results seem to be available for these cases.
>
> We also explain the connection between our capacity assumptions and the mentioned work on heavy-tailed empirical risk minimization and the entropy conditions [37-38] in more detail for the non-expert, as the eigenvalue decay in our work is directly linked to the entropy numbers [3].
> Please also refer to our reply to **Reviewer rSYV** for the extension of our results to other loss functions.
>
> **Practical implications and usage of squared loss**
>
> While our work is of course of theoretical nature, we fully agree that the practitioner requires a presentation of the main insights created by our work. We add a paragraph summing these insights up:
>
> - squared loss is often not used in practice when one expects heavy-tailed noise as it is (i) sensitive to outliers when used *without regularization* (see related work section in the manuscript), and (ii) the population loss is generally undefined when the response only admits moments of order $q<2$ ("very-heavy-tailed")
>
> - however, *with regularization* and in the setting $q \geq 3$, we show that it is (i) asymptotically optimal with the same regularization schedule as in the light-tailed setting, (ii) the achievable rates with optimal regularization are the fastest available as far as we are aware, (iii) in a *large sample setting*, the confidence behaviour is essentially subgaussian and allows to treat the problem like a light-tailed problem, (iv) in a *small sample setting*, stronger regularization is to be performed due to the impact of outliers.
>
> We want to stress that deriving empirical parameter choice rules and more practical insights are very important and can be investigated in future work based on the theoretical foundation we aim to provide; see below.
>
> **Practical guidelines for choosing $\alpha$ (empirical parameter choice rules)**
>
> We agree that the practitioner requires consistent empirical choices of regularization parameters.
> We see our current work as the theoretical foundation and a starting point for the derivation of these results. In fact, analogously to similar work for the light-tailed case, empirical parameter choice rules can be derived for example based on the so-called *discrepancy principle* (see e.g. [4,5]). Our upper bounds allow to insert various parameter choices rules constructed from data in order to investigate their efficiency. It is now of importance to understand how heavy-tailed noise affects the estimation of these parameter choices, potentially again by making use of a Fuk-Nagaev inequality.
>
> **References**
>
> [1] H. Wen, A. Betken, and W. Koolen. On the robustness of kernel ridge regression using the Cauchy loss function, arxiv.org/abs/2503.20120 (2025).
>
> [2] [1] I. Steinwart and A. Christmann. Support Vector Machines (2008).
>
> [3] I. Steinwart, D.R. Hush, C. Scovel. Optimal rates for regularized least squares regression. COLT (2009).
>
> [4] H. W. Engl, M. Hanke, A. Neubauer. Regularization of Inverse Problems (1996).
>
> [5] A. Celisse, M. Wahl. Analyzing the discrepancy principle for kernelized spectral filter learning algorithms. JMLR (2021).

---

> > ### Comment · Reviewer_JkXL · 2025-08-05
> >
> > Thank you for your detailed response, I would keep my score.

---

### Official Review · Reviewer_AM9J · 2025-07-05

**Clarity:** 3
**Significance:** 3
**Originality:** 3
**Rating:** 4
**Confidence:** 3

**Summary:**

The paper analyzes the behavior of ridge regression in reproducing kernel Hilbert spaces when the noise is heavy-tailed (only finite qth moment for q>= 3). They discuss the excess risk bound by using vectorized version of standard Fuk-Nagaev inequality used for heavy tails. Further the authors claim that ridge regression achieves minimax optimal sample complexity. Hence, as compared to existing methods, the paper is giving theoretical guarantees for minimax optimal results for heavy tailed noise which was assumed to be sub-exponential before.

**Questions:**

1) Is the computational complexity affected by heavy tailed noise? If yes, what is the specific relation?
2) What will be the performance of other regularization methods such as Lasso for heavy tailed noise? Can the minimax results be claimed for other regularization methods?
3) In Assumption 2.2, the authors assume q >= 3. Can the analysis be extended for q >=2? The authors discuss existing works in line 93 to have results for q > 0.
4) Can the authors clarify their contributions as compared to [51]? In line 94, the authors say that [51] follows a similar approach.

**Ethical Concerns:**

["NO or VERY MINOR ethics concerns only"]

**Final Justification:**

Overall, this is a good paper. I mentioned the strengths of the paper. Most of my comments/questions or potential weaknesses were addressed by the authors in the rebuttal phase. The authors also did the required empirical study, which makes the paper strong. Hence, I recommend acceptance of the paper.

**Limitations:**

yes

**Quality:**

3

**Strengths And Weaknesses:**

## Strengths:
- Achieving the minimax optimal guarantees
- Generalizing the assumption to heavy tailed noise instead of sub-exponential  distribution.
- The generalized extension of Fuk-Nagaev inequality with decomposition of tail behavior into sub-Gaussian and polynomial term is novel. It can be used for other works.

## Weakness:
- There is no empirical evidence of the proposed theoretical claims.
- Heavy tailed noise can be motivated more in the introduction section.
- What specific kernels follow assumption 2.1 or can you examples of kernels which do not follow assumption 2.1? Do the polynomial kernels in https://proceedings.neurips.cc/paper_files/paper/2015/file/f7f580e11d00a75814d2ded41fe8e8fe-Paper.pdf follow Assumption 2.1?

---

> ### Author Rebuttal · Authors · 2025-07-28
>
> We thank the reviewer for their positive feedback and constructive comments and questions. We have addressed all points of criticism and made appropriate edits to the manuscript. We elaborate on the details below.
> We would be grateful if the reviewer might consider increasing their score in case they are satisfied with the proposed changes.
>
> **Empirical evaluations**
>
> We agree with the reviewer that empirical experiments improve the quality of the paper.
> In fact, just after the submission, we completed a series of numerical evaluations which exactly confirm the behavior described by our theory. We intend to include these experiments in the final paper using the additional space.
>
> We simulate iid $(X,Y)$ data pairs with $Y = f_\star(X) + \varepsilon$ with fixed $f_\star$ in a Gaussian RKHS and consider a heavy tailed case in wich $\varepsilon$ is $t$-distributed, and a light-tailed case in which $\varepsilon$ is normal distributed (both with the same variance).
>
> For several parameter configurations (distributional assumptions, regularization parameters, number of samples), we compute $\hat{f}_\alpha$ across series of realizations with different random seeds for both noise types and approximate the $L^2$-excess risk
> $
> \lVert \hat{f}\_{\alpha} - f\_{\star}\rVert
> $
> by sampling from $\pi$.
>
> Given the different realizations for all parameter configurations, we compute the empirical distribution of the excess risks and approximate the quantile function
> $$
>     Q(1-\delta) :=
> \inf_t \left(
>     P\\left[ \left\lVert \hat{f}\_{\alpha} - f\_{\star} \right\rVert  \leq t \right]
>  \geq 1-\delta \right).
> $$
>
> As reflected by our theory, the quantile behaviour of both noise models for larger $\delta$ overlaps almost exactly (subgaussian confidence regime). For small $\delta$ however, the heavy-tailed model exhibits much larger error quantiles than the Gaussian model (polynomial confidence regime). The transition threshold between these two cases is affected by the number of samples and the regularization parameter choice exactly as described by our theory.
>
> **Highlighting motivation to consider heavy-tailed noise**
>
> We agree. We added a paragraph to the introduction that highlights the practical relevance and applications of heavy-tailed noise models (finance/actuarial science [7], communication networks, biology and atmospherical sciences [8]) as well as the investigation of the robustness of ridge regression when standard noise models are violated.
>
> **Assumption 2.1 and relevant kernels**
>
> Assumption 2.1 is quite standard in the sense that it is fundamental in
> classical work for kernel ridge regression that we build upon [3] and subsequent analyses in the literature (see e.g. [2] and references therein).
> In particular, it applies to commonly used continuous radial kernels on $R^d$ such as the Gaussian kernel and the Matèrn kernel. However, it is noteworthy that it only applies to the mentioned polynomial kernels when the input domain is bounded (or more generally when the distribution of $X$ has compact support). For unbounded covariate distributions, the boundedness assumption is *generally violated for polynomial kernels*. We added a remark to the paper that clarifies the applicability of Assumption 2.1.
>
> **Computational complexity**
>
> The computational complexity of the ridge regression estimate is *entirely unaffected by the distribution of the data*, in particular by the assumption of heavy-tailed noise.
> It is dictated by a convex program in dual form [1, Example 11.8] which is often solved by inverting the regularized kernel matrix [3, Prop. 8]. We added a sentence highlighting the numerical evaluation of the empirical estimate after line 156 for completeness.
>
> **Other regularization methods**
>
> Currently, the approach only applies to the Tikhonov regularization method. We are working on a direct extension to the family of *spectral regularization methods*, which includes for example truncated SVD and Landweber iteration (which translates to a fixed learning rate gradient descent in this case) [9]. This requires some modifications of our proofs, but allows to work in the same framework. The LASSO penalty seems to require more work and a slightly adjusted framework, but this is definitely an interesting direction for future research.
>
> **Extension of the results to $q \leq 3$**
>
> We believe this is an important point. The answer is split into two distinct cases.
>
> *First case:* $2 \leq q \leq 3$. The real-valued Fuk-Nagaev inequality is valid for $q >2$. We hence believe that our Hilbert space version uses suboptimal truncation arguments and can be extended to $q>2$ by adopting more precise arguments [5] to the infinite-dimensional setting. The remaining parts of our proofs (i.e. balancing the bias and variance) allow for the case $q>2$, leading directly to the desired results in if the concentration bound can be generalized. We further believe that the value of the constants $c_1$ and $c_2$ can be made more precise. We are currently working on this.
>
> *Second case:* $q < 2$. When $\varepsilon$ does not admit a second moment, the response $Y = f_\star(X) + \varepsilon$ does also not necessarily admit a second moment and the squared loss is not guaranteed to be well defined. In such a case, our approach and the Fuk-Nagaev inequality would not apply and other losses such as the Huber loss or Cauchy loss would be required to ensure finiteness of the loss, leading to a different type of analysis [4]. In this setting however, exact optimality is still unclear as far as we are aware. Also note that our rates are faster than the rates in [4] whenever $q>3$.
>
> We added a “Extensions and Limitations” paragraph to end of the introduction to highlight these facts for the reader.
>
> **Technical comparison to Zhu et al. (2022)**
>
> The authors of the above paper use a *real-valued* Fuk-Nagaev inequality to bound the complexity of stochastic gradient descent for ordinary (e.g. linear and unregularized) least squares in finite dimensions. The only technical similarity to our paper is the application of a Fuk-Nagaev type inequality to provide tail estimates for the noise.
>
> In particular, our technique differs from their analysis in the following ways:
>
> (i) We combine concentration arguments with operator perturbation theory in the context of a linear inverse problem, while Zhu et al. perform an analysis of the SGD iteration via a martingale decomposition.
>
> (ii) Zhu et al. operate in a strongly convex setting with a nonzero lower bound on the smallest eigenvalue of the covariance matrix (see their Assumption 2). Our covariance operator is compact and infinite-dimensional and hence not allowing for such a bound. Consequently, we need to express the hardness of the learning problem as a source condition to obtain rates.
>
> (iii) We derive minimax optimal rates while Zhu et al. focus on the general sample complexity and the transition between subgaussian and polynomial confidence, not explicitly discussing details related to optimality.
>
> **References**
>
> [1] I. Steinwart and A. Christmann. Support Vector Machines (2008).
>
> [2] S. Fischer and I. Steinwart. Sobolev norm learning rates for regularized least-squares
> algorithms. JMLR (2020).
>
> [3]  F. Cucker and S. Smale. On the Mathematical Foundations of Learning. Bulletin of the AMS (2001).
>
> [4] H. Wen, A. Betken, and W. Koolen. On the robustness of kernel ridge regression
> using the Cauchy loss function, arxiv.org/abs/2503.20120 (2025).
>
> [5] E. Rio. About the constants in the Fuk-Nagaev inequalities. Electronic Communications in Probability (2017).
>
> [6] V. Yurinsky. Sums and Gaussian vectors (1995).
>
> [7] P. Embrechts, C. Klüppelberg, and T. Mikosch. Modelling extremal events: for insurance and finance (1997).
>
> [8] J. Nair, A. Wierman, and B. Zwart. The fundamentals of heavy tails. Properties,
> emergence, and estimation (2022).
>
> [9] H. W. Engl, M. Hanke, A. Neubauer. Regularization of Inverse Problems (1996).

---

> > ### Comment · Reviewer_AM9J · 2025-08-02
> >
> > I thank the authors for their detailed response and references. I would also like to thank them for doing experiments. I intend to retain my score rating.

---

### Note · Authors · 2025-08-15

Dear AC, dear reviewers,

Thank you for contributing to the process of reviewing this paper.

We are happy to see that all reviewers recommend an acceptance of our paper and agree that our contribution is novel and meaningful. We have addressed all questions, and the reviewers are satisfied with our responses. The final manuscript will be updated, and all discussed points are addressed accordingly.

As a central addition, we provide the results of numerical experiments (see detailed description in the response to Reviewer AM9J), which precisely confirm the behavior of the distribution of the excess risk characterized by our theoretical results.

Furthermore, we are excited that the importance of our work is highlighted by multiple *potential future* directions outlined in the exchange with the reviewers. For example, our approach can be used to investigate unbounded kernels with finite higher moments, empirical regularization parameter choices, and more general spectral regularization schemes. We expect more general results for learning with heavy-tailed noise to be within reach by adapting our technique.

Kind regards,

The authors

---

### Decision · Program_Chairs · 2025-09-17

**Decision:**

Accept (spotlight)

**Comment:**

This work derives excess risk bounds of ridge kernel regression with Tikhonov regularization in a heavy-tailed noise setting with bounded first three moments. Notably, the analysis shows that in the large sample regime, the convergence rate behaves similarly in this heavy-tailed setting and in the light-tailed setting with bounded or subexponential noise. This implies a certain level of robustness of square loss to heavy-tailed noise, despite it not being a predominant choice to learn from heavy-tailed data.

The provided analysis constitutes a significant advance from the previous results restricted to subexponential noise. The technical approach using Fuk-Nagaev inequality has the potential of being applied to future works, some of which were outlined during discussion with the reviewers and would be interesting to include in the final version. It was also suggested to provide a discussion on the practical implications of the analysis, to which the authors have responded with some remarks on the use of square loss in heavy-tailed noise setting and the choice of regularization parameter. A shared concern among the reviewers is the lack of empirical validation, in response to which the authors claimed to have obtained numerical results that match the behavior predicted by the analysis.